# RhoA- and Cdc42-induced antagonistic forces underlie symmetry breaking and spindle rotation in mouse oocytes

**Benoit Dehapiot**[1,2]*, **Raphaël Clément**[1], **Anne Bourdais**[2], **Virginie Carrière**[2], **Sébastien Huet**[2], **Guillaume Halet**[2]*

**1** Aix Marseille Université, CNRS, IBDM-UMR7288, Turing Center for Living Systems, Marseille, France,
**2** Univ Rennes, CNRS, IGDR—UMR 6290, Rennes, France

* benoit.dehapiot@univ-amu.fr (BD); guillaume.halet@univ-rennes1.fr (GH)

## Abstract

Mammalian oocyte meiotic divisions are highly asymmetric and produce a large haploid gamete and 2 small polar bodies. This relies on the ability of the cell to break symmetry and position its spindle close to the cortex before anaphase occurs. In metaphase II–arrested mouse oocytes, the spindle is actively maintained close and parallel to the cortex, until fertilization triggers sister chromatid segregation and the rotation of the spindle. The latter must indeed reorient perpendicular to the cortex to enable cytokinesis ring closure at the base of the polar body. However, the mechanisms underlying symmetry breaking and spindle rotation have remained elusive. In this study, we show that spindle rotation results from 2 antagonistic forces. First, an inward contraction of the cytokinesis furrow dependent on RhoA signaling, and second, an outward attraction exerted on both sets of chromatids by a Ran/Cdc42-dependent polarization of the actomyosin cortex. By combining live segmentation and tracking with numerical modeling, we demonstrate that this configuration becomes unstable as the ingression progresses. This leads to spontaneous symmetry breaking, which implies that neither the rotation direction nor the set of chromatids that eventually gets discarded are biologically predetermined.

## Introduction

Meiosis is the evolutionarily conserved cell division process by which haploid gametes are generated from diploid germ cells. Because meiosis is universally initiated after a premeiotic S phase, 2 consecutive rounds of cell division—meiosis I and meiosis II—are required to generate gametes with only one copy of each chromosome. In the mammalian female germline, meiotic divisions are highly asymmetric, producing a large oocyte and 2 smaller polar bodies. This enables the oocyte to conserve most of its cytoplasmic resources (for instance, mRNA, mitochondria, and ribosomes) while discarding supernumerary genetic material before parental genome fusion. The asymmetry of oocyte meiotic divisions relies on the ability of the cell to break symmetry and position its spindle close to the cortex before anaphase occurs. The

**Data Availability Statement:** All relevant data are within the paper and its supporting information files.

**Funding:** The authors have received funding from the following sources: - Research grant ATIP from

the Centre National de la Recherche Scientifique/ CNRS (2009; to GH): https://insb.cnrs.fr/fr/atip-avenir - Research grant from the Ligue Contre le Cancer (2015; to GH): https://www.ligue-cancer. net/article/27236_appels-projets-recherche - Academic scholarship from the Society for Reproduction and Fertility (2013; to GH): https:// srf-reproduction.org/grants-awards/grants/ academic-scholarship-fund/ - PhD scholarship (3 years) from the french Ministry of Research and Higher Education (to BD): https://www. enseignementsup-recherche.gouv.fr/ - PhD scholarship (6 months) from the Fondation pour la Recherche Médicale/FRM (to BD): https://www. frm.org/chercheurs/appel-a-projets-frm The funders had no role in study design, data collection and analysis, decision to publish, or preparation of the manuscript.

**Competing interests:** The authors have declared that no competing interests exist.

**Abbreviations:** EDM, Euclidean distance map; F-actin, actin filament; PB2, second polar body; PIV, particle image velocimetry; PLK1, Polo-like kinase 1; ROCK1, Rho-associated protein kinase 1; RT, room temperature; sPB, small polar body; $t_i$ rotation, initial time of rotation.

spindle is indeed responsible for setting up the cleavage plane [1] and must therefore be off-centered to allow an uneven partitioning of the gamete. In mitotic cells, spindle positioning is primarily achieved through interactions with the cell cortex, which exerts pulling forces on centrosome-nucleated astral microtubules projecting from the spindle poles [2,3]. Intriguingly, oocytes from most species are devoid of centrioles and therefore lack genuine astral microtubules [4]. Hence, oocytes must rely on alternate strategies to position their spindle.

Over the last 2 decades, seminal studies have shed light on how actin filaments (F-actin) play a pivotal role in positioning the spindle during both mouse oocyte meiotic divisions [5–7]. During early metaphase I, the spindle assembles in the central region of the oocyte and slowly relocates toward the nearest cortical region [8]. Spindle migration relies on a cytoplasmic network of filaments, assembled by the Formin-2 and Spire1/2 nucleators [9–12]. This network indeed supports both pushing and pulling forces that are necessary to relocate the spindle [13–17]. Once fully migrated, the spindle enters in anaphase I, and half of the homologous chromosomes are discarded into the first polar body. The ovulated oocyte then enters into the prolonged metaphase II arrest, during which the gamete actively maintains its spindle off-centered and parallel to the cortex until fertilization occurs. The metaphase II spindle positioning also depends on the actin cytoskeleton and the emergence of a positive feedback between chromosome positioning and cortical polarization. Indeed, it has been shown that chromatin generates a gradient of active Ran GTPase (RanGTP), which triggers polarization of the cortex in a dose- and distance-dependent manner [18,19]. This polarization is characterized by an accumulation of F-actin, referred to as the F-actin cap, encircled by a ring of activated myosin-II [19]. Remarkably, the F-actin cap is, in turn, capable of attracting chromosomes in the cortical vicinity and thus consolidates the polarity by locally increasing the concentration of RanGTP. This can be explained by the ability of the F-actin cap to generate a polarized flow of filaments. Indeed, by doing so, F-actin promotes the emergence of lateral streaming of cytoplasmic materials, which converge in the center of the gamete, before pushing back the spindle against the polarized cortex [14,20]. A similar directional pushing force was suggested to accelerate spindle migration in meiosis I, once maternal chromosomes are within approximately 25 μm from the cortex [14]. While the molecular details underlying the cortical actomyosin polarization are not yet fully elucidated, we and others have shown that F-actin cap formation requires the polarized activation of the Cdc42 GTPase (Cdc42GTP) and the Arp2/3 complex, downstream of RanGTP [19–26]. Inhibiting Ran GTPase, by overexpressing a dominant negative form (RanT24N), leads to a complete loss of the cortical actomyosin polarity and an absence of cytoplasmic streaming [19,20].

The second meiotic division is triggered by sperm entry and leads to the segregation of the sister chromatids (later referred to as the DNA clusters) between the fertilized oocyte and second polar body (PB2). The success of this division relies on the ability of the spindle, which lies parallel to the cortex during metaphase II, to reposition itself perpendicular to the cell periphery to allow cytokinesis ring closure at the base of the polar body [27]. As originally described by Maro and colleagues, this process is accompanied by the reorganization of the oocyte cortex with 2 F-actin–rich protrusions emerging above the spindle, before one of them becomes the PB2 while the other one retracts in the later stage of division [28]. While evidence suggests that spindle rotation is prevented by actin depolymerization, myosin II inhibition or RhoA inactivation [29–31], the underpinnings of this unique symmetry breaking event have long remained elusive. In a recent study, Wang and colleagues provided new insights into this mechanism, showing that rotation is achieved through an asymmetric distribution of forces along the anaphase II spindle [32]. It was suggested that spindle rotation arises from cytoplasmic flows of opposite directions at the 2 ends of the spindle, consecutive to spontaneous symmetry breaking in the distribution of Arp2/3 and myosin-II at the cortex [32].

In the present study, we showed that spindle rotation results from 2 antagonistic forces. First, a RhoA-dependent ingression of the cytokinetic furrow in the central spindle region, and second, a Ran/Cdc42-dependent attraction of DNA clusters to the polarized cortex. Using numerical modeling, we have noted that the anaphase II spindle configuration becomes unstable as ingression progresses. The slightest initial asymmetry leads to symmetry breaking and spindle rotation. We developed a live 3D segmentation and tracking procedure to confirm these views in vivo and showed that, for instance, the early positioning of the DNA clusters relative to the cortex biases the orientation of spindle rotation. Overall, our data suggest that the rotation of the spindle is the result of a stochastic and spontaneous symmetry breaking process.

## Results

### Stochastic symmetry breaking underlies spindle rotation

To monitor spindle rotation in living oocytes, we injected metaphase II–arrested oocytes with cRNAs encoding H2B-mCherry and eGFP-MAP4, to respectively label chromosomes and spindle microtubules. After a 3-h culture period to allow protein expression, we triggered meiosis II resumption by incubating oocytes in an ethanol-complemented culture media [22]. This rapidly induced parthenogenetic activation of the gamete, which proceeds to PB2 emission. We imaged oocytes at 37˚C, using confocal laser scanning microscopy (see selected example in Fig 1A, S1 (brightfield) and S2 Movies (microtubule), and overall procedure in Methods).

At anaphase onset, segregated chromatids started to move toward the spindle poles, forming 2 DNA clusters (see 3 min, Fig 1B). Simultaneously, the cortical region facing the central spindle invaginated and 2 distinct membrane protrusions assembled above each of the DNA clusters (see 3 to 15 min, Fig 1B). This was rapidly followed by the rotation of the anaphase II spindle, such that one of the DNA clusters was extruded in the nascent polar body, while the other one remained within the oocyte (see 20 to 30 min, Fig 1B). Remarkably, the cortical protrusion overlaying the internalized cluster collapsed in the latter stage of the spindle rotation (see 35 to 45 min, Fig 1B). Eventually, the second meiotic division ended with the closure of the cytokinetic ring at the base of the polar body (see 60 min, Fig 1A).

Using particle image velocimetry (PIV; see Methods), we showed that the triggering of anaphase was also accompanied with dramatic changes in the cytoplasmic streaming pattern (Fig 1C and S3 Movie). Indeed, the postactivation flows were inverted as compared to those observed in metaphase II oocytes [20], with cytoplasmic material flowing away from the polarized domain and moving toward the center of the gamete (see t = 15 min, S1A Fig and S4 Movie). The peak of flow reversal clearly coincided with the cortical invagination occurring at the central spindle region (see S1B–S1D Fig) and was consistently observed in the control oocyte population (S1E and S1F Fig). Remarkably, the collapse of the cortical protrusion (see t = 40 min, S1A Fig) and cytokinesis ring closure (see t = 55 min, S1A Fig) also correlated in time with the emergence of distinct flows in the latter stages of rotation.

We next devised an automated 3D segmentation and tracking procedure to monitor over time the position of the DNA clusters within the volume of the oocyte (S2A and S2B Fig, S5 Movie, and Methods). Using this method, we measured the distance between the 2 DNA clusters and linearly fitted the early time points in order to achieve time registration of all recordings (Fig 1D and Methods). We also measured the spindle rotation angle (as defined in S2B Fig) and used logistic curve fit to extract monotonic smoothed profiles. This allowed us to define the initial time of rotation (hereafter referred to as $t_i$ rotation), corresponding to the time when the rotation reaches 5% of the total fitted rotation (Fig 1E and Methods). By looking

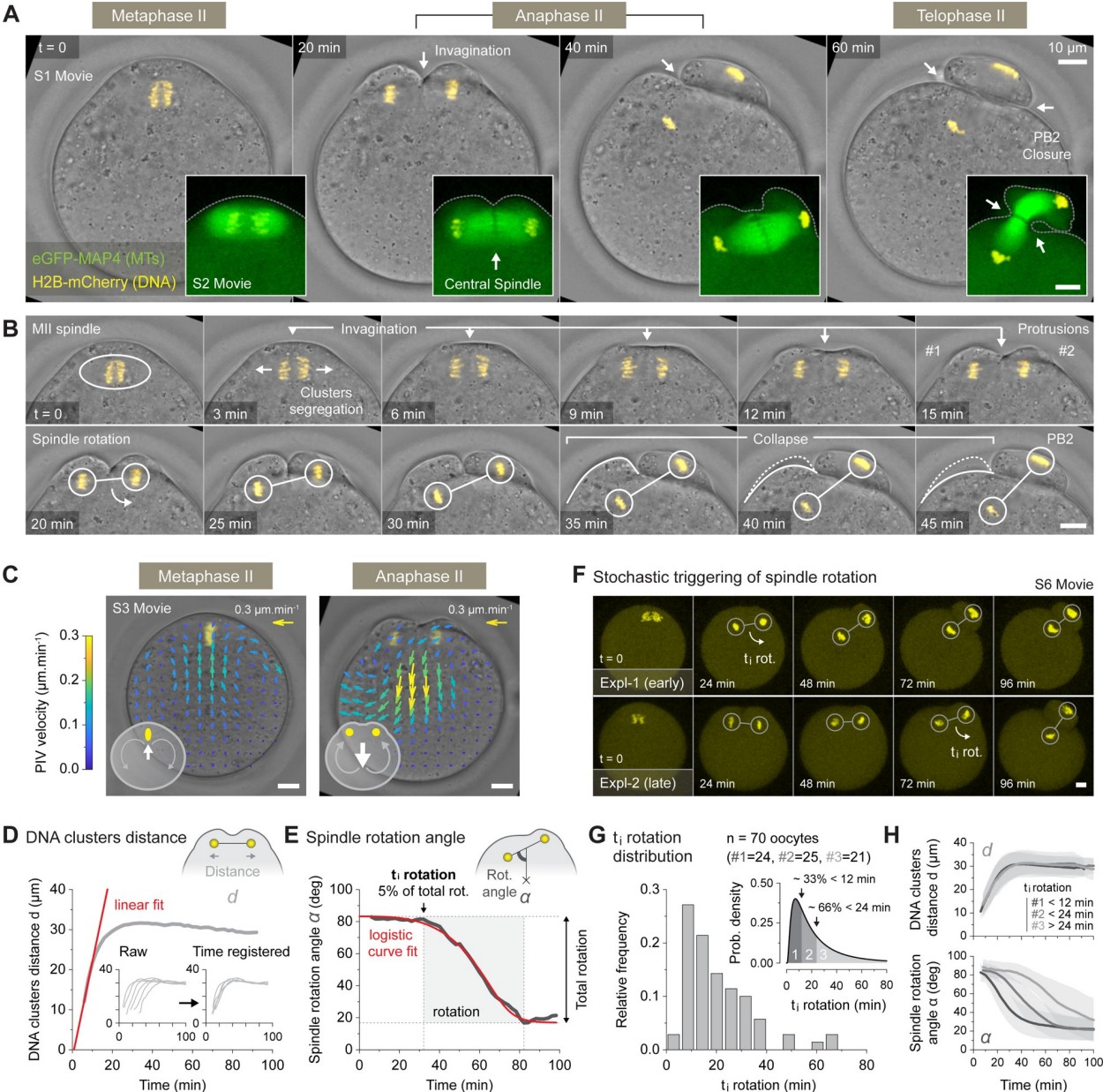

**Fig 1. Stochastic symmetry breaking underlies spindle rotation.** (A) Live imaging of activated metaphase II oocytes undergoing spindle rotation. The oocytes were injected either with the H2B-mCherry DNA marker alone (main brightfield time series, S1 Movie) or in combination with the eGFP-MAP4 MT marker to follow spindle dynamics (inserted spindle time series, S2 Movie). The white arrows show membrane invaginations leading to PB2 closure. (B) High temporal resolution montage of the main brightfield time series shown in panel A, where the key events of the second meiotic division are highlighted. (C) PIV measurements showing the inversion of cytoplasmic streaming occurring during the transition between metaphase II and anaphase II (S3 Movie). The vector field color code indicates the speed of tracked particles. The top-right arrows show size and color of a 0.3-μm.min-1 vector. (D) Variation over time of DNA cluster distance $d$ in a selected oocyte (main graph). The red line shows the linear fit used for time registration of the oocyte population (inserted graphs). (E) Variation over time of the spindle rotation angle $\alpha$ in a selected oocyte. The red curve shows the logistic fit used to extract rotation parameters. The $t_i$ rotation is defined as the time when the rotation reaches 5% of the total fitted rotation. (F) Two examples of H2B-mCherry expressing oocytes illustrating the stochastic triggering of spindle rotation (see S6 Movie). The white arrows indicate the approximate start of the rotation. (G) Frequency distribution of $t_i$ rotation for a control oocyte population ($n$ = 70; 9 independent experiments) (main graph). PDF of the $t_i$ rotation distribution used to determine 3 equiprobable categories of increasing $t_i$ rotation (inserted graph). (H) Variation over time ± SD of the DNA cluster distance $d$ (top graph) and spindle rotation angle $\alpha$ (bottom graph) averaged per category as defined in panel G. Data underlying this figure can be found in S1 Data. Scale bars = 10 μm. MT, microtuble; PB2, second polar body; PDF, probability density function; PIV, particle image velocimetry; $t_i$ rotation, initial time of rotation.

at various examples (Fig 1F and S6 Movie) or more specifically, at the distribution of the $t_i$ rotation (Fig 1G), we observed that the oocytes did not trigger spindle rotation synchronously with regard to the DNA cluster separation (used here as our time reference). This was particularly visible when splitting the oocyte population into 3 categories, according to their $t_i$ rotation (see Fig 1G). In doing so, one can easily observe that, while distance curves align perfectly, the rotation curves are shifted in time between categories (Fig 1H). This high degree of variability indicates that the symmetry breaking process is rather stochastic and that the dynamics of rotation is not tightly controlled. This led us to further investigate the underlying mechanisms of symmetry breaking.

## The central spindle/RhoA pathway is required for spindle rotation

The RhoA GTPase is a conserved regulator of cytokinesis in animal cells, through the assembly of a contractile cleavage furrow in the cortical region overlying the central spindle [33–35]. Accordingly, RhoA activity is required for polar body cytokinesis in mouse oocytes [31,36,37]. The canonical RhoA activation pathway involves the guanine nucleotide exchange factor ECT2 [38–40], which is recruited to the central spindle by the so-called centralspindlin complex (MKLP1, MgcRacGAP) [41–43]. The recruitment of ECT2 relies on the phosphorylation of the MgcRacGAP subunit by Polo-like kinase 1 (PLK1) [44,45]. This pathway is recapitulated in the context of the mouse oocyte in the diagram of Fig 2A.

We performed immunostaining against RhoA (see Methods) on differentially staged oocytes and succeeded to monitor the GTPase localization throughout the second meiotic division. Our experiments revealed that, while showing no particular localization during metaphase II arrest, RhoA strongly accumulated in the cortical region facing the central spindle during anaphase II (Fig 2B). This accumulation was associated with the recruitment of other classical cytokinesis factor such as ECT2, the scaffolding protein anillin, and the Rho-associated protein kinase 1 (ROCK1) (S3A Fig). The RhoA GTPase assembled a hemi-ring invaginating between the 2 cortical protrusions overlaying the separating DNA clusters (see z-projections, Fig 2B). The hemi-ring remained open until the end of spindle rotation, after which it closed around the central spindle at the base of the polar body (see telophase II, Fig 2B).

We next sought to inhibit the central spindle/RhoA pathway to examine its contribution to spindle rotation. Thus, we treated oocytes with Bi-2536, a small molecule inhibitor reported to specifically target PLK1 in somatic cells [46] and mouse oocytes [47]. As expected, PLK1 inhibition prevented the recruitment of ECT2 to the central spindle while preserving the localization of its binding partner MgcRacGAP (S3B Fig). As a result, both the cortical accumulation of RhoA and the central cleavage furrow ingression were abolished in this condition (Fig 2C). This led to a sharp reduction of the PB2 extrusion rate (controls $n = 68/81$ approximately 85% PB2, Bi-2536 $n = 3/105$ approximately 3% PB2) (Fig 2D).

When monitored live, the Bi-2536–treated oocytes demonstrated chromatid cluster separation but failed to rotate their spindles. Surprisingly, some of these spindles ($n = 10/35$ approximately 29%) rather engaged in a spin around the cell periphery before stopping in the latter stages of division (Fig 2E, S3C Fig, and S7 Movie). In terms of rotation, around the central spindle axis, this translated in a clear difference between conditions, with spindles in controls undergoing an approximately 50° angle variation while the treated ones remained almost parallel to the cortex (mean angle after 120 min, controls 33.7° ± 14.1 SD, Bi-2536 78.0° ± 10.3 SD) (Fig 2F and 2G). Interestingly, we also noted that cytoplasmic streaming inversion was no longer visible in treated oocytes, which suggest that, indeed, the contraction of the cytokinetic furrow is responsible for this phenomenon (Fig 2H). Instead, the Bi-2536–treated oocytes

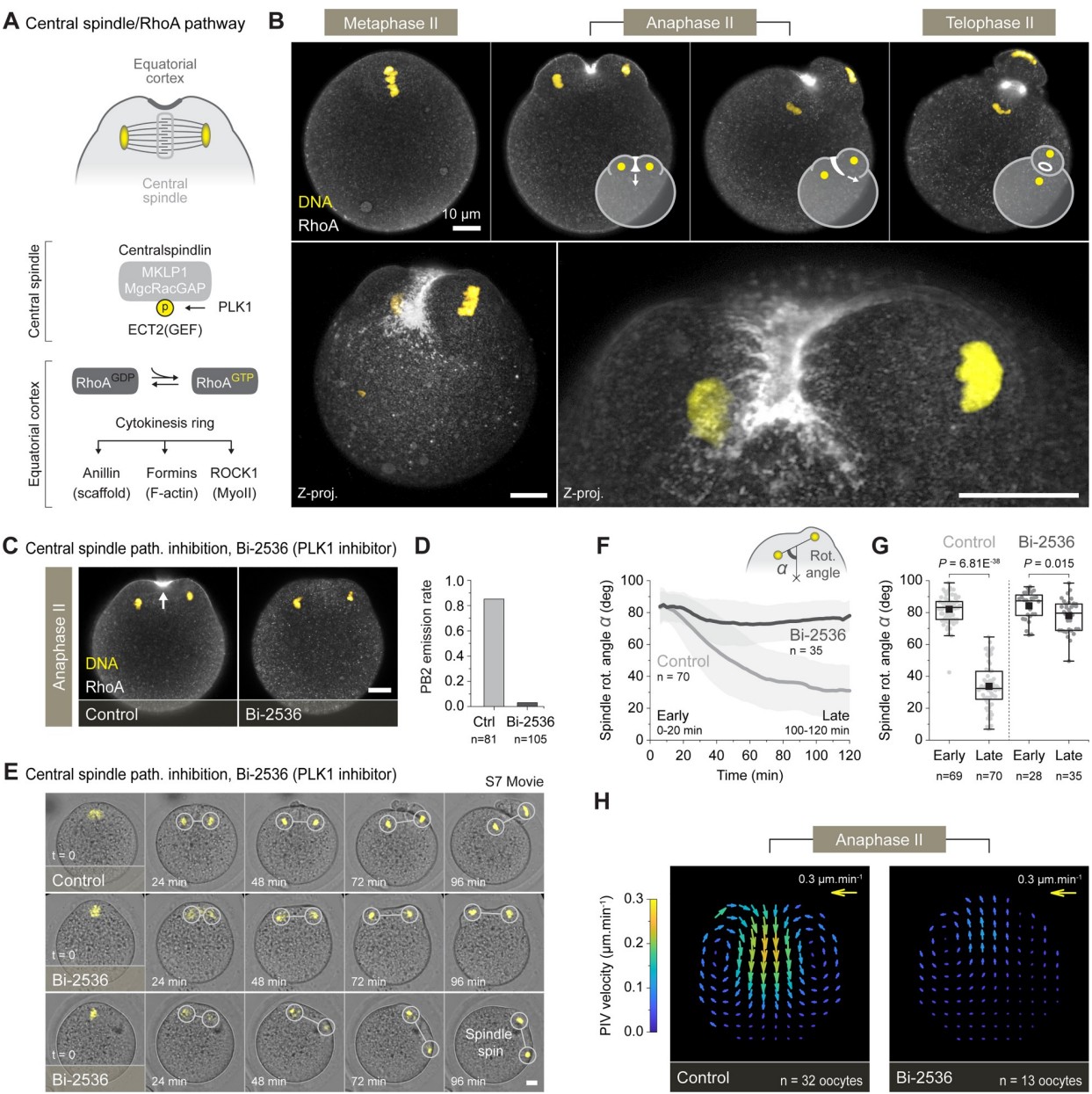

**Fig 2. The central spindle/RhoA pathway is required for spindle rotation.** (A) Simplified diagram of the central spindle pathway leading to the activation of the RhoA GTPase and the assembly of the cytokinesis ring. (B) Fixed RhoA (immunostaining) and DNA (To-Pro-3) imaging of differentially staged oocytes undergoing spindle rotation. The top panel shows single confocal planes with superimposed diagrams illustrating the progressive closure of the cytokinesis ring. The bottom panels represent maximum intensity z-projections of multiple confocal planes to better visualize the hemi-ring distribution of RhoA. (C) Inhibiting the central spindle pathway, using the Bi-2536 (PLK1 inhibitor), prevented the recruitment of RhoA and the cortical ingression (see white arrow). Fluorescence patterns shown in C were consistently observed in 33 control and 25 Bi-2536–treated oocytes, both gathered from 3 independent experiments. (D) PB2 emission rate in 81 control and 105 Bi-2536–treated oocytes, both gathered from 4 independent experiments. (E) Live imaging of activated control and Bi-2536–treated oocytes (S7 Movie). The oocytes were injected with the H2B-mCherry DNA marker. Among the Bi-2536–treated oocytes, about one-third ($n = 10/35$ approximately 29%) of the spindles engaged in a spin around the cell periphery, while the others ($n = 25/35$ approximately 71%) simply remained parallel to the cortex. (F) Averaged variation over time ± SD of the spindle rotation angle $\alpha$ in control and Bi-2536–treated oocytes. (G) Box plots showing the distribution of early (0–20 min) and late (100–120 min) spindle rotation angle $\alpha$ in control and Bi-2536–treated oocytes. Measurements in F and G were performed on 70 control and 35 Bi-2536–treated oocytes, gathered respectively from 9 and 6 independent experiments. (H) PIV vector fields showing the averaged cytoplasmic streaming pattern in 32 control and 13 Bi-2536–treated oocytes, both gathered from 5 independent experiments. Box plots in G extend from the first (Q1) to the third (Q3) quartile (where Q3–Q1 is the IQR); whiskers are Q1 or Q3 ± 1.5 × IQR; horizontal lines represent the median; and black squares represent the mean. *p*-values in G were obtained using a two-sided Mann–Whitney test. Data underlying this figure can be found in S1 Data. Scale bars = 10 μm. IQR, interquartile range; PB2, second polar body; PIV, particle image velocimetry; PLK1, Polo-like kinase 1; ROCK1, Rho-associated protein kinase 1.

rather displayed metaphase-like flows with cytoplasmic material moving toward the polarized cortex.

Overall, these data indicate that the central spindle/RhoA pathway, which promotes cytokinetic ring assembly and cortical ingression, is required to drive the spindle rotation. Interestingly, upon RhoA inhibition, the spindle still initiates a directional, yet unproductive, spinning movement around the oocyte. This suggests that additional mechanisms may contribute to the symmetry breaking.

## Cortical actomyosin polarization reorganizes during anaphase II

As mentioned in the introduction, the chromosomes carried by the metaphase II spindle generate a gradient of RanGTP that triggers the polarization of the actomyosin cortex in a dose- and distance-dependent manner [18,19]. Remarkably, the polarized cortex is, in turn, capable of attracting the chromosomes to the cell periphery, allowing the oocyte to keep its spindle off-centered during metaphase II arrest [20]. This ability to modulate the spindle localization encouraged us to investigate whether the DNA-induced polarity pathway still operates during anaphase II and could therefore be involved in the symmetry breaking occurring during spindle rotation.

Consistent with previous observations [18], metaphase II oocytes exhibited a polarized F-actin cap surrounded by a ring of activated myosin-II in the cortical region overlying the spindle (see top panel, Fig 3A). In cortical fluorescence intensity profiles (see Methods), this took the shape of a broad F-actin peak flanked by 2 smaller myosin-II peaks corresponding to opposite sides of the ring (Fig 3B). This pattern persisted after the anaphase onset but separated in 2 smaller caps/rings overlaying each of the segregating DNA clusters (see middle panels, Fig 3A and 3B). However, in the latter stages of anaphase II, an asymmetry in cortical myosin-II distribution became evident. Indeed, while the ring configuration was maintained in the nascent polar body, myosin-II invaded the polarized cortex overlying the internalized DNA cluster (see bottom panels, Fig 3A and 3B). As a result, both the F-actin and myosin-II formed a cap in the collapsing protrusion. Importantly, another cortical actomyosin subpopulation emerged as a patch in front of the central spindle during anaphase II (arrowheads in middle panels of Fig 3A and 3B). This second subpopulation likely belongs to the cytokinesis ring since it colocalized with active RhoA (see Methods, S4A and S4B Fig). This was confirmed using PLK1 (Bi-2536) and ROCK1 (Y-27632) inhibitors, which both prevented the actomyosin patch formation (S4C and S4D Fig). Note that RhoA activation persisted in latrunculin A–treated oocytes, further suggesting that RhoA lies upstream of the F-actin patch (S4C and S4D Fig). Interestingly, the F-actin cap and myosin-II rings overlaying the DNA clusters were not affected by PLK1 and ROCK1 inhibitors in metaphase II or anaphase II (S4E and S4F Fig). This demonstrates that these 2 cortical actomyosin subpopulations are differentially regulated.

We next quantified the average cortical F-actin and myosin-II levels as a function of the distance between the cortex and DNA clusters in fixed oocytes (see Methods and Fig 3C and 3D). In both metaphase II and anaphase II oocytes, the cortical F-actin intensity decreased as the distance between the cortex and DNA clusters increased. In contrast, and in line with its ring localization, cortical myosin-II reached its maximum intensity at a certain distance from the DNA clusters (19.9 μm ± 1.6 SD in metaphase II and 13.5 μm ± 1.8 SD in anaphase II, see Fig 3D). This indicates that, like in metaphase II, cortical actomyosin polarization is primarily regulated by the proximity of the chromatin in activated oocytes. At close range, RanGTP is high and cortical F-actin is strongly enriched. With distance, F-actin levels decrease and myosin-II assembles as a ring in the distal part of the gradient (top panel, Fig 3E). Halving the amount of DNA in the anaphase II clusters, due to sister chromatid separation, does not alter this pattern

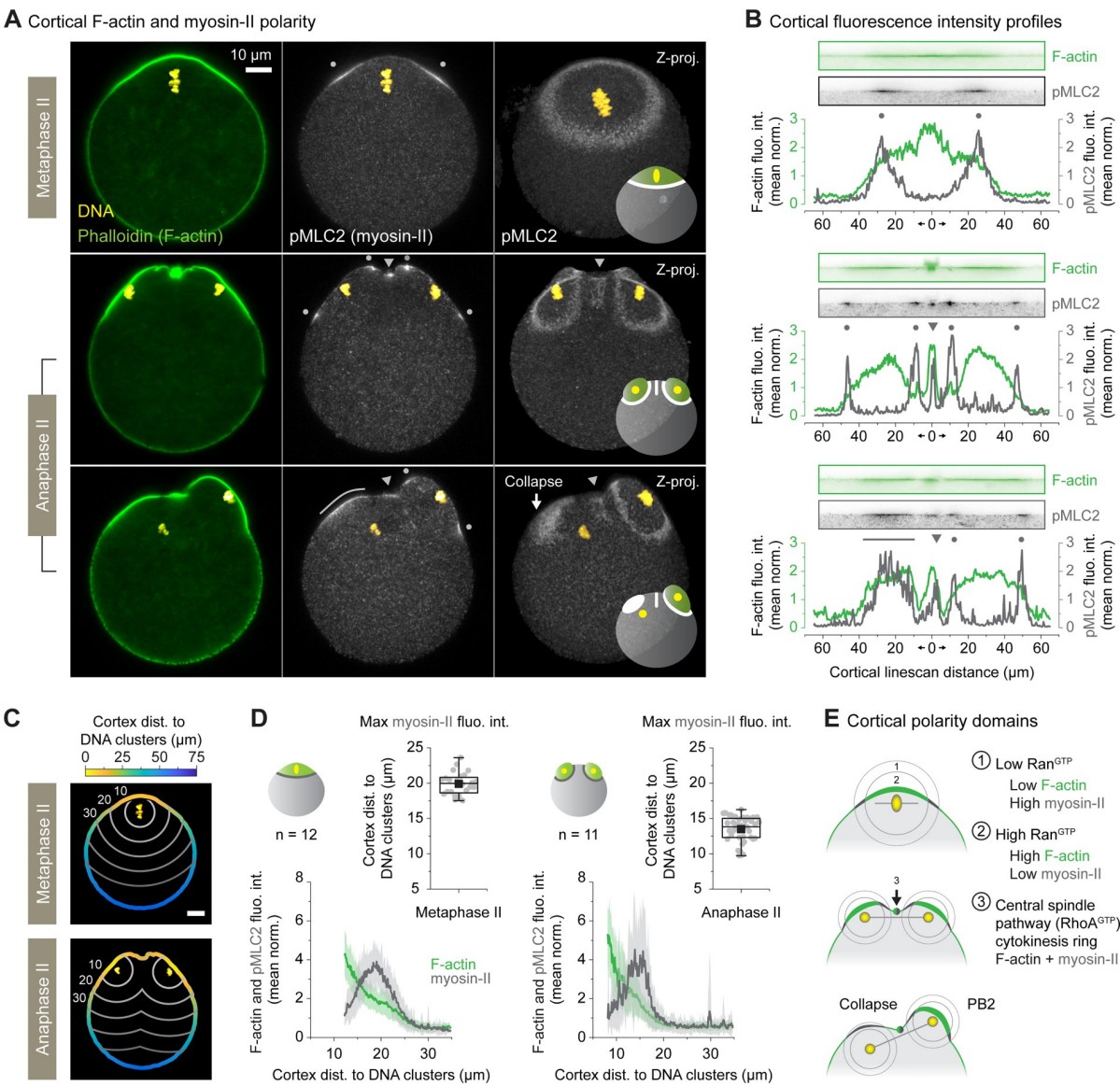

**Fig 3. Cortical actomyosin polarization reorganizes during anaphase II.** (A) Fixed F-actin (phalloidin staining), myosin-II (pMLC2 immunostaining), and DNA (To-Pro-3) imaging of differentially staged oocytes undergoing spindle rotation. The 2 left columns of images show single confocal planes, while the rightmost column shows maximum intensity z-projections of multiple confocal planes. The gray dots and arrowheads indicate, respectively, the DNA-induced and the central spindle-induced cortical myosin-II subpopulation. The white arrow shows the collapsing cortical protrusion. The superimposed diagrams in the rightmost column summarize the observed cortical actomyosin localization. (B) Cortical fluorescence intensity profiles recapitulating F-actin and myosin-II polarization patterns observed in A. The fluorescence intensities are normalized to the mean of each profile. The gray dots and arrowheads show the DNA- and the central spindle-induced cortical myosin-II populations, respectively (as in A). Straightened cortical images used to extract the fluorescence intensity profiles are shown above the graphs. (C) Diagrams showing the color-coded cortex distance to DNA clusters in both metaphase II (top) and anaphase II oocytes. The gray lines delineate oocyte regions equidistant from the DNA clusters. (D) Line graphs: averaged ± SD cortical F-actin (phalloidin staining) and myosin-II (pMLC2 immunostaining) fluorescence intensities as a function of the cortex distance to DNA clusters. The fluorescence intensities are normalized to the mean of each profile. Box plots: distribution of the cortical myosin-II peaks distance to the DNA clusters (see Methods). The quantifications were performed on 12 metaphase II and 11 anaphase II oocytes, gathered from 2 independent experiments. (E) Diagrams illustrating how the RanGTP gradient, centered on the chromosomes, shapes cortical actomyosin polarization during the second meiotic division. Cortical F-actin accumulates in the proximal part of the gradient (zone 2), while myosin-II concentrates in the periphery (zone 1). Also note the cortical enrichment of actomyosin due to the cytokinesis ring (zone 3). Parameters for box plots in D are as described in Fig 2. Data underlying this figure can be found in S1 Data. Scale bars = 10 μm. F-actin, actin filament; PB2, second polar body; pMLC2, phosphorylated myosin light chain 2.

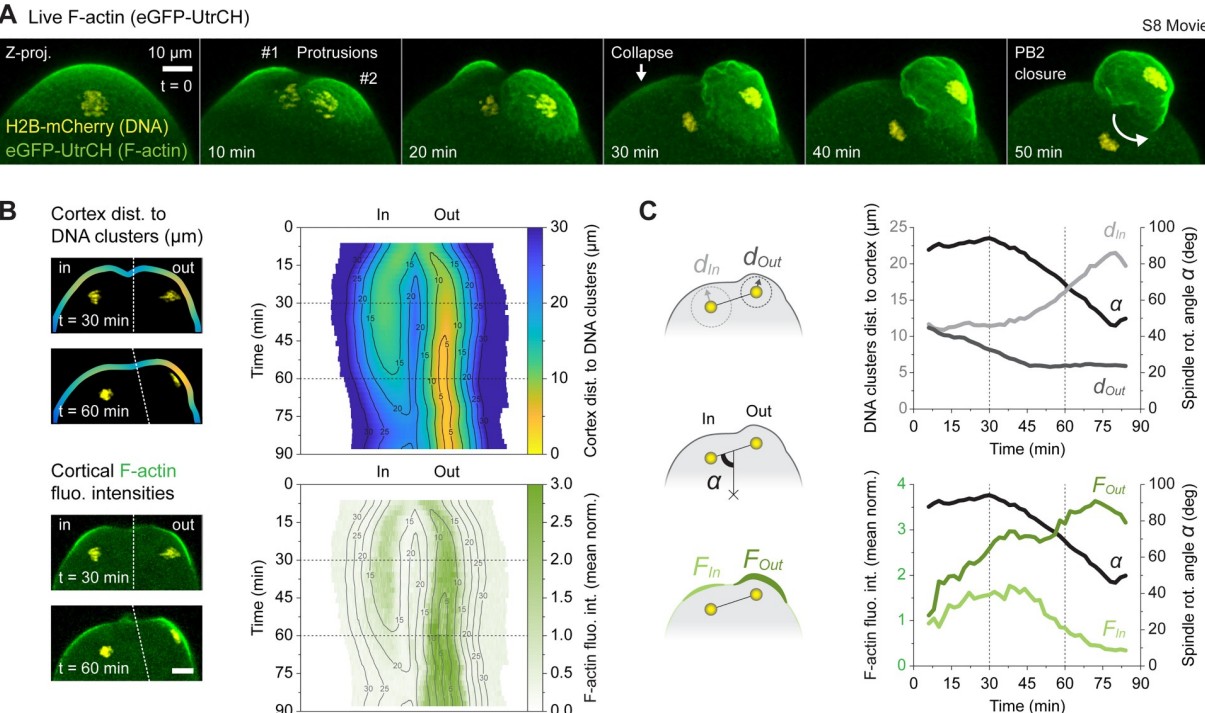

**Fig 4. DNA cluster positioning governs cortical F-actin polarization.** (A) Live imaging of an activated oocyte undergoing spindle rotation. The oocyte was injected with a combination of the H2B-mCherry DNA marker and the eGFP-UtrCH F-actin marker (S8 Movie). The images show maximum intensity z-projections of multiple confocal planes where the key events of the second meiotic division are highlighted. (B) 2D maps of the polarized domain showing the variation over time of the cortex distance to DNA clusters (color-coded top panel) and the cortical F-actin levels (bottom panel). The maps are extracted from images as shown on the left panel. The isodistance lines are used as landmarks to delimit region of the cortex equidistant from the DNA clusters. The dashed lines delineate the best focus plane for each DNA cluster. (C) 1D representation of the data shown in B (see Methods). The internalized DNA cluster and the extruded DNA cluster are designated as *in* and *out*, respectively. The graphs show variation over time of the DNA cluster distance to the cortex (top panel, $d_{In}$ and $d_{Out}$) and the cortical F-actin levels in each polarized domain (bottom panel, $F_{In}$ and $F_{Out}$). The variation of spindle rotation angle $\alpha$ is also represented on both graphs. Corresponding diagrams are shown on the left panel. Data underlying this figure can be found in S1 Data. Scale bars = 10 μm. F-actin, actin filament; PB2, second polar body.

but only reduces the size of the polarization domains (middle panel, Fig 3E). The later recruitment of myosin-II in the collapsing cortical protrusion is consistent with these observations. Indeed, as the spindle rotates, the distance between the cortex and the internalized set of chromatids increases. As a consequence, only a low dose of RanGTP reaches the cortex at this stage, which promotes the cortical recruitment of myosin-II (bottom panel, Fig 3E).

We pursued our investigations by coinjecting oocytes with cRNAs encoding the H2B-mCherry DNA marker and the eGFP-UtrCH F-actin probe [48]. This allowed us to monitor live cortical F-actin dynamics throughout the division process. Our movies recapitulated what we observed in fixed experiments, with 2 polarized F-actin caps emerging above the segregating sets of chromatids, one of which becoming the PB2 while the other one collapsed in the later stages of division (Fig 4A and S8 Movie). We next sought to correlate, over time, the cortical level of F-actin with the distance separating the DNA clusters from the cortex. To do so, we generated color-coded maps of the polarized domain, displaying the spatiotemporal variations of these 2 quantities (see Methods and Fig 4B). We also determined restricted cortical domains, overlaying the inward and the outward clusters, to average the results and produce 1D profiles (see Methods and Fig 4C). Using both methods, we observed a clear correlation between the proximity of the DNA clusters and the level of cortical F-actin polarity. We also

noticed, as will be discussed later, that the outward cortical domain tended to be more polarized (closer to the DNA cluster and presenting more F-actin) at the onset of spindle rotation (see time = 30 min in Fig 4C).

Altogether, the above findings demonstrate that the DNA-induced polarity pathway, operating during metaphase II, is still at play after anaphase onset. What differs in anaphase II is that the polarized cortex divides in 2 smaller domains, above each of the segregating DNA clusters. The polarization intensity of the subdomains evolves during the rotation and depends on the proximity of the chromatin.

## The polarized cortex exerts attraction forces on chromatid clusters

We next wanted to know if the polarized anaphase II cortex exerts attractive forces on chromosomes, as it does in late metaphase I [14] and metaphase II [20]. We found striking evidence supporting this hypothesis by treating oocytes with a low dose of nocodazole while they were already engaged in spindle rotation (Fig 5A and S9 Movie). Indeed, by doing so, we were able to slightly disrupt spindle integrity and occasionally trigger the detachment of DNA clusters from the meiotic apparatus. When the internalized cluster detached, it systematically moved

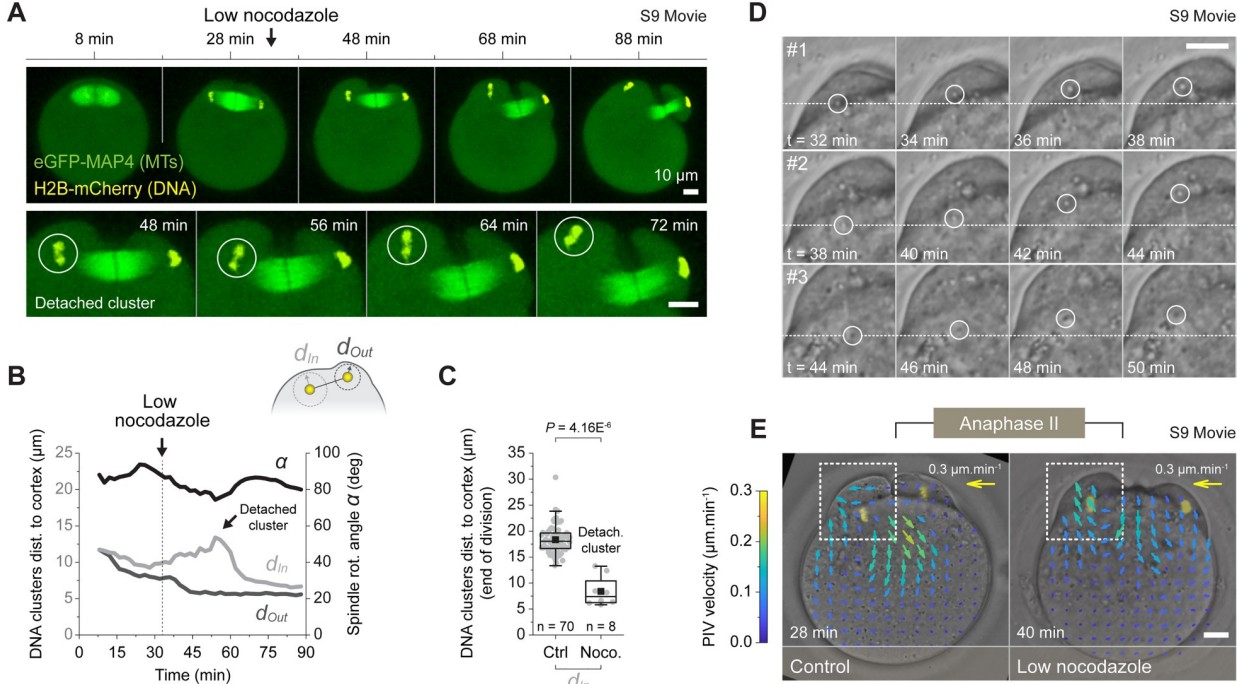

**Fig 5. Nocodazole-induced DNA cluster detachment reveals cortical attraction forces.** (A) Live imaging of an activated oocyte undergoing spindle rotation while being treated with a low dose of nocodazole (supplemented to the culture medium at time = 34 min). The oocyte was injected with a combination of the H2B-mCherry DNA marker and the eGFP-MAP4 MT marker (top panel and S9 Movie). The bottom panel shows a magnification to highlight the detachment of the internalized DNA cluster from the spindle and its migration toward the overlying cortical protrusion (see white circles). (B) Graph showing the variation over time of the DNA cluster distance to the cortex ($d_{In}$ and $d_{Out}$) and the rotation angle $\alpha$ of the nocodazole-treated oocyte shown in A. (C) Box plots showing the distribution of the internalized DNA cluster distance to the cortex ($d_{In}$) at the end of the second meiotic division in 70 control and 8 nocodazole-treated oocytes whose internalized cluster has detached from the spindle. These oocytes were gathered from 9 and 4 independent experiments, respectively. (D) brightfield time-lapse sequences, extracted from the nocodazole-treated oocyte shown in A (see S9 Movie), showing examples of manually tracked cytoplasmic particles moving toward the cortical protrusion (see particles in white circles and dotted lines as spatial references). (E) PIV measurements, performed on a control and the nocodazole-treated oocyte shown in A, showing local metaphase-like flows directed toward the cortical protrusion. Parameters for box plots and statistics in C are as described in Fig 2. *p*-value in C was obtained using a two-sided Mann–Whitney test. Data underlying this figure can be found in S1 Data. Scale bars = 10 μm. MT, microtubule; PIV, particle image velocimetry.

back toward the cortical protrusion in a highly directional manner (see bottom panel in Fig 5A). This resulted in a sharp decrease of the distance between the internalized cluster and the cortex (see $d_{In}$ in Fig 5B), which was found to be significantly reduced, as compare to controls, at the end of division (Fig 5C). Interestingly, we also noticed that along with the internalized cluster, many cytoplasmic particles were attracted to the polarized cortex. To illustrate this phenomenon, we manually tracked some of these particles to reveal that a local flow was established toward the cortical protrusion (see Fig 5D and S9 Movie). This local flow could also be detected in PIV analysis, particularly when inverted cytoplasmic streaming were not too predominant (see low nocodazole in Fig 5E). Importantly, control oocytes also displayed similarly oriented flows above the internalized DNA clusters (see control in Fig 5E). Altogether, these results demonstrate that DNA clusters still undergo attractive forces from the polarized cortex after anaphase onset. These forces most likely result from local outward-oriented metaphase-like flows that persist despite the general reversal of cytoplasmic streaming due to the cytokinetic contraction.

As a next step, we wanted to prevent these metaphase-like flows to see how interfering with cortical attractive forces might influence spindle rotation. To this end, we sought to inhibit cortical F-actin polarity, since the latter has been described as responsible for the emergence of these flows in both metaphase I and II [14,20]. To achieve this inhibition, we targeted the Ran signaling pathway (see diagram, Fig 6A) by overexpressing dominant negative forms of Ran GTPase itself or its downstream effector Cdc42 (respectively, RanT24N [49] and Cdc42T17N [50]). Cdc42 is known to promote cortical F-actin polarization in metaphase II oocytes through the activation of N-WASP and the Arp2/3 complex [20–24]. Accordingly, expression of either dominant negative forms resulted in a complete loss of the F-actin and myosin-II cortical polarity in metaphase II oocytes (S5A and S5B Fig).

Using the same strategy, we next examined the effects of Cdc42 inhibition in activated oocytes. We did not consider inhibiting the upstream Ran GTPase, as RanGTP plays an additional role in maintaining spindle integrity [18,51]. Thus, we acquired confocal time series of activated oocytes expressing Cdc42T17N and H2B-mCherry, and we observed 2 distinct phenotypes (Fig 6B and S10 Movie). In the first group of oocytes (*n* = 19/27 approximately 70%), the anaphase spindle remained off-centered (spindle-centroid distance > 20 μm) and a PB2 protruded, though smaller in size than in controls. In the second group (*n* = 8/27 approximately 30%), the anaphase spindle relocated substantially toward the oocyte center (spindle-centroid distance < 20 μm), and PB2 was not emitted (see Methods and Fig 6C). As overexpressing Cdc42T17N alone is known to inhibit PB2 emission in a large majority of activated oocytes [22], we considered the small polar body (sPB) phenotype as an incomplete Cdc42 inhibition. We were supported in this idea by targeting the downstream Arp2/3 nucleator (see diagram, Fig 6A) using the CK-666 inhibitor. Indeed, our experiments revealed that spindles of CK-666–treated oocytes were similarly relocating to the center of the cell (Fig 6D). This validated the spindle relocation phenotype observed upon Cdc42 inhibition and further indicated that this effect was due to the cortical F-actin polarization activity of the GTPase.

We also noted that, alike control oocytes, Cdc42T17N-injected oocytes underwent a reversal of cytoplasmic streaming after anaphase onset (Fig 6B and 6F and S10 Movie). This observation prompted us to investigate whether this flow inversion could be responsible for spindle relocation upon Cdc42 inhibition. Indeed, this would further suggest that Cdc42 and the cortical F-actin polarization generate attractive forces that are necessary to counteract the reverted cytokinetic flows. To address this question, we measured cytoplasmic streaming velocity using PIV (see Methods and S1 Fig for the procedure) and compared it to the speed at which the meiotic spindle relocated in Cdc42T17N oocytes (see PIV measurements in S5C Fig and linear fits in S5E Fig). We observed a clear correlation between these 2 parameters, with spindle-

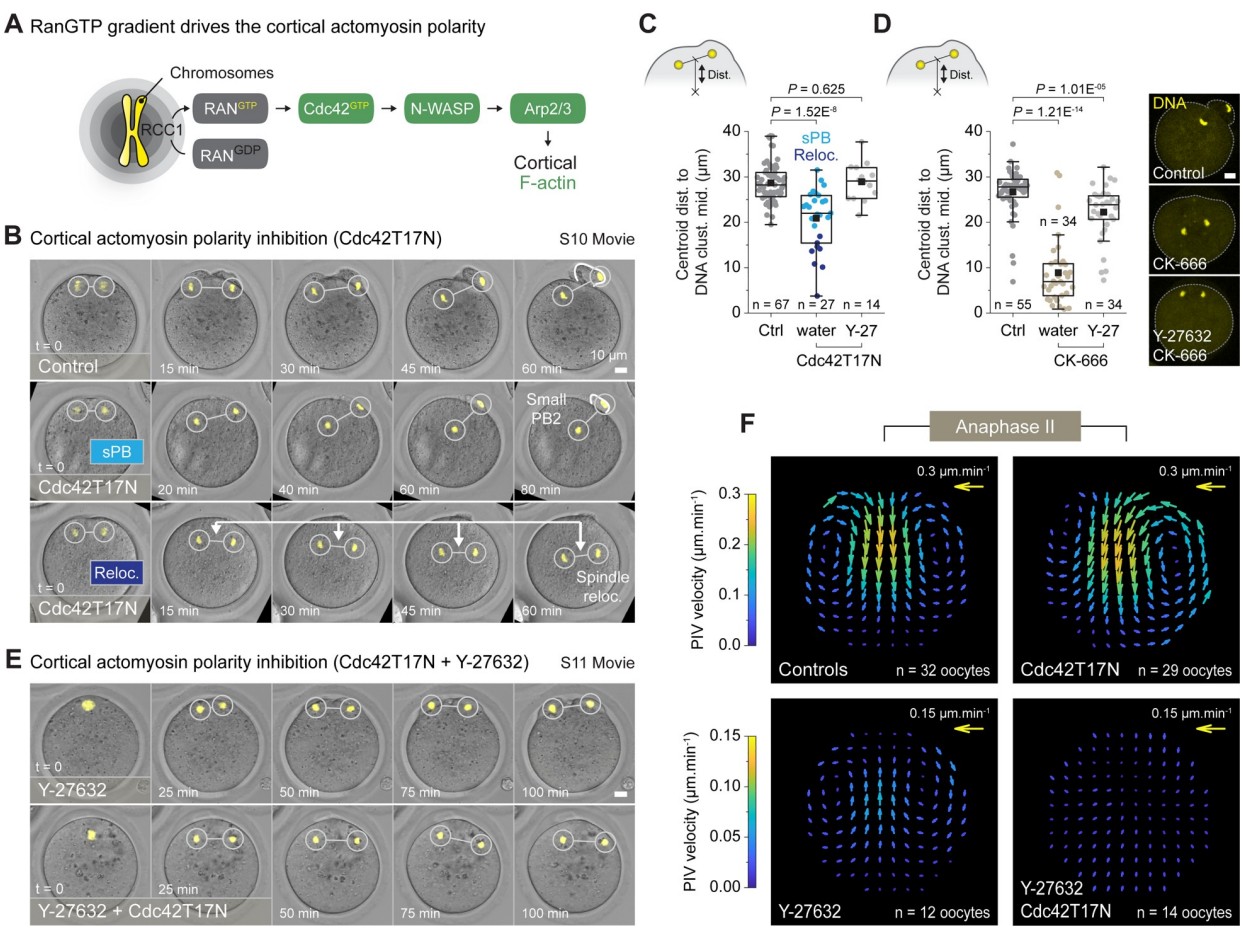

**Fig 6. Cdc42 promotes cortical attraction forces through F-actin polarization.** (A) Simplified diagram of the Ran/Cdc42 signaling, emanating from the chromosomes, and leading to the cortical accumulation of F-actin through the activation of the Cdc42, N-WASP, and Arp2/3 pathway. (B) Live imaging of activated oocytes injected with the H2B-mCherry DNA marker alone (control) or in combination with Cdc42T17N (S10 Movie). The Cdc42T17N-injected oocytes presented 2 characteristic phenotypes: a first group extruding a small PB2 (light blue sPB), as compared to controls, and a second group whose spindle relocated toward the center of the cell (dark blue Reloc). (C) Box plots showing the distribution of the distance from the oocyte centroid to the DNA cluster midpoint 90 min after activation. Measurements were performed on 67 control, 27 Cdc42T17N, and 14 Cdc42T17N + Y-2762–treated oocytes, gathered, respectively, from 9, 3, and 4 independent experiments. (D) Box plots showing the distribution of the distance from the oocyte centroid to the DNA cluster midpoint 90 min after activation. Measurements were performed on 55 control, 34 CK-666–treated, and 34 CK-666 + Y-27632–treated oocytes, gathered, respectively, from 8, 3, and 2 independent experiments. Images (right panel): selected examples showing the localization of DNA clusters in fixed oocytes from the different experimental conditions. (E) Live imaging of Y-27632–treated oocytes (activated), injected with the H2B-mCherry DNA marker alone (control) or in combination with Cdc42T17N (S11 Movie). (F) PIV vector fields showing the averaged cytoplasmic streaming pattern in 32 control, 29 Cdc42T17N, 12 Y-27632–treated, and 14 Cdc42T17N + Y-27632–treated oocytes, gathered, respectively, from 8, 3, 3, and 4 independent experiments. Parameters for box plots in C and D are as described in Fig 2. $p$-values in C and D were obtained using a two-sided Mann–Whitney test. Data underlying this figure can be found in S1 Data. Scale bars = 10 μm. F-actin, actin filament; PB2, second polar body; PIV, particle image velocimetry; sPB, small polar body.

relocating oocytes undergoing, on average, stronger reverse flows (reloc. 0.31 μm.min$^{-1}$ ± 0.07 SD) than oocytes extruding an sPB (0.21 μm.min$^{-1}$ ± 0.08 SD) (S5D, S5F and S5G Fig). However, this increased flow velocity alone cannot explain the spindle relocation phenotype. Indeed, many control oocytes displayed flow velocities as high as those measured in spindle-relocating oocytes. This indicates that other parameters, such as a lack of attractive forces, are involved in this phenotype.

Finally, to confirm a causal relationship between spindle relocation and inverted cytoplasmic streaming, we prevented cytokinetic contraction by treating oocytes with the ROCK1

inhibitor, Y-27632. As expected, this treatment resulted in a complete loss of cytoplasmic streaming in Cdc42T17N oocytes (Fig 6E and 6F and S11 Movie) and rescued the spindle relocation phenotype in both Cdc42T17N and CK-666–treated oocytes (Fig 6C and 6D). Interestingly, when applied alone on control oocytes, Y-27632 also prevented the reversed cytoplasmic streaming, but not the metaphase-like flows that were still observable (Fig 6E and 6F and S11 Movie). This again showed that metaphase-like flows persist during anaphase II and further revealed that these flows result from Cdc42 activity.

Overall, our nocodazole and Cdc42 inhibition experiments demonstrated that the polarized cortex still exerts attractive forces on chromosomes during anaphase II. These forces are necessary to maintain the spindle in the cell periphery during the division and emerge, most likely, from metaphase-like flows resulting from Cdc42 activity and its ability to polarize the F-actin cortex. However, due to the spindle relocation phenotype observed upon Cdc42 inhibition, we were not able to draw conclusions about the potential involvement of these attractive forces in the symmetry breaking. We therefore devised a numerical model that we confronted with our experimental results to provide further insights into this particular aspect.

## Antagonistic forces yield symmetry breaking and spindle rotation

Based on our findings, we considered that the anaphase II spindle is mainly subject to 2 antagonistic forces: (1) the inward contraction of the cytokinesis ring occurring in the central-spindle region; and (2) the outward attraction of the DNA clusters to their respective polarized cortical domain. To translate this in mathematical terms, we devised a simplified model in which a semiflexible spindle connects 2 DNA clusters encapsulated in a circular oocyte (see Methods and Fig 7A). To simulate the cytokinetic furrow ingression, we prescribed a constant inward velocity to the spindle midpoint, oriented perpendicular to the spindle axis. On the other hand, we modeled the attraction of DNA clusters using an effective potential, akin to a soft-core potential (see the potential profile in Fig 7B), accounting for a feedback such that the attraction of the clusters increases as their distance to the cortex decreases. Note that the use of an effective potential, added to the fact that physical quantities (forces, viscosity, etc.) are not known quantitatively, makes the model essentially qualitative and presented in arbitrary units. Its purpose is thus mainly to outline the fundamental mechanisms of symmetry breaking.

As the ingression progresses, the attraction domains and the spindle midpoint are no longer aligned, which prevents DNA clusters from remaining both apposed to the cortex. In the absence of numerical noise, the simulation remained perfectly symmetric and the spindle was simply pushed, parallel to the cortex, inside the oocyte (top panel, Fig 7C and S12 Movie). However, this configuration was actually unstable, and the addition of a small Gaussian noise to the clusters position (see Methods) was enough to break the symmetry and systematically trigger spindle rotation (middle and bottom panels, Fig 7C and S12 Movie). Rotation resulted in a more energetically favorable configuration, with one of the DNA clusters returning to its potential well, while the other eventually escaped from it (see arrows in Fig 7B). Note that many sources of noise are possible in the biological system. These may include (but are not limited to) an asymmetry of the initial configuration, an asymmetry of the attraction forces, thermal or active fluctuations of the clusters' position, and an imperfect centering of the central spindle or of the ingression. Here, for the sake of simplicity, we only considered noise on the clusters' position, although any source of noise would lead to symmetry breaking.

We next analyzed the dynamics of symmetry breaking in simulations and observed that, as seen in experiments, the distribution of the $t_i$ rotation (defined as in Fig 1E) was spread out over time and was skewed to the right (Fig 7D). We used this temporal difference to group our real and simulated oocyte populations into 3 categories, according to how soon they break

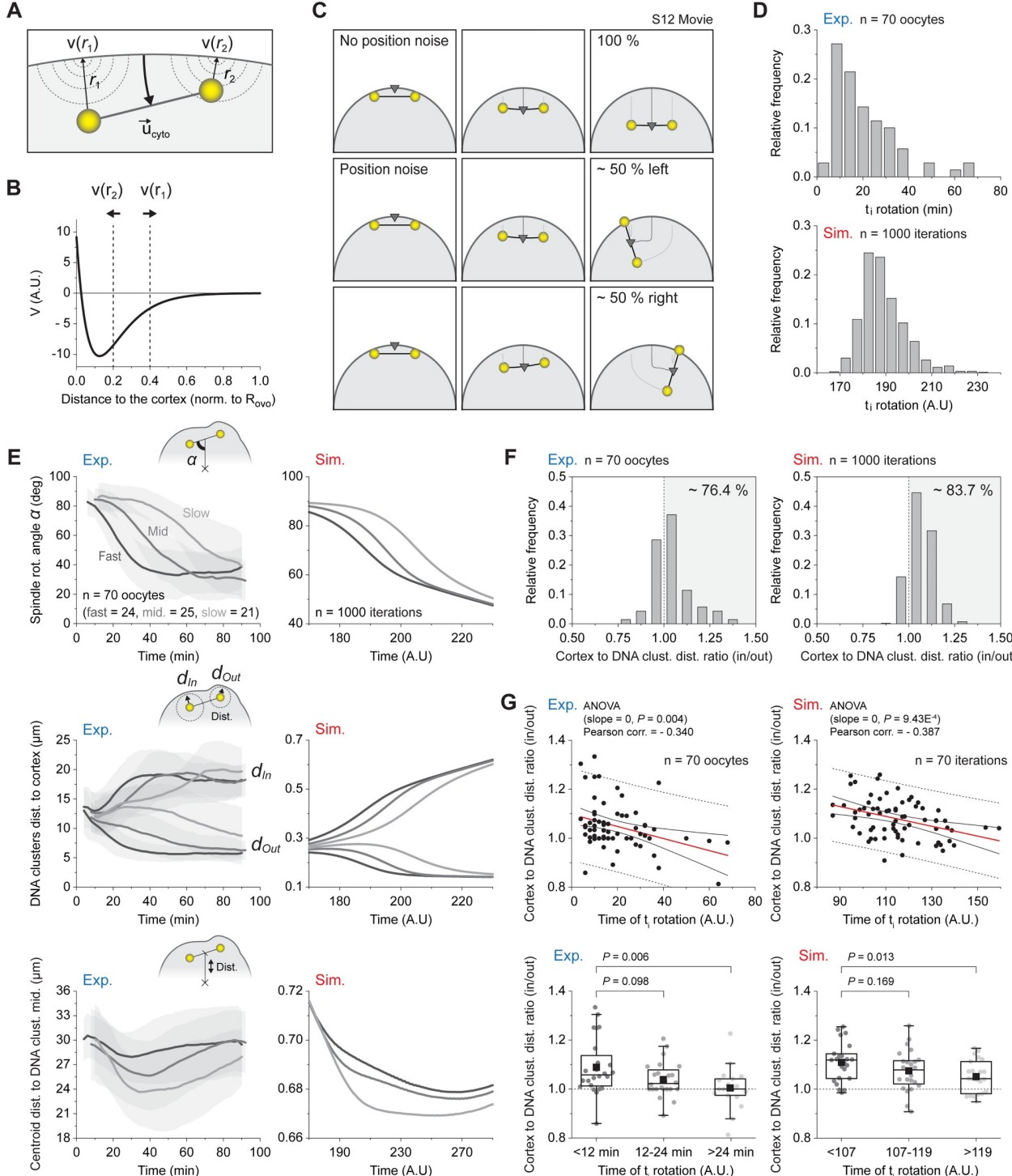

**Fig 7. Antagonistic forces yield symmetry breaking and spindle rotation.** (A) Diagram recapitulating the main elements used in the numerical model. A constant inward velocity, $\vec{u}_{cyto}$, was applied to the spindle midpoint to recapitulate the cytokinetic furrow ingression. The attraction of the DNA clusters for the cortex was modeled through an effective potential, $V(r)$. The strength of this potential decays exponentially as the DNA cluster distance to the cortex increases (see $r_1$ and $r_2$). See Methods for further explanation. (B) Shape of the effective potential, $V(r)$, modeling the attraction of the DNA clusters for the cortex. (C) Time-lapse sequences of typical simulations with (top row) or without (middle and bottom rows) Gaussian noise added to the DNA cluster position (S12 Movie). (D) Experimental (top) and simulated (bottom) distributions of the initial rotation times ($t_i$ rotation). (E) Experimental (left column) and simulated (right column) variation over time ± SD of the spindle rotation angle $\alpha$ (top row), the DNA cluster distance to the cortex (middle row), and the distance between the oocyte centroid and the DNA cluster midpoint (bottom row).

For each graph, the oocyte population was separated into 3 categories of increasing $t_i$ rotation (see Fig 1H). (F) Experimental (left) and simulated (right) distributions of the averaged DNA cluster distance to cortex ratio (in/out) in the first 10 min of recordings. Values >1 (see gray background) indicate that the extruded cluster (out) was closer to the cortex before the initiation of rotation. (G) Experimental (left column) and simulated (right column) DNA cluster distance to cortex ratio (in/out) versus $t_i$ rotation, shown as scatter plots (top row) or binned box plots (bottom row). The red line in the scatter plots represents a linear fit of the point cloud. The plain black lines delimit the 95% confidence band, while the dashed lines delimit the 95% prediction band. The experimental results in D–G were obtained from a population of 70 control oocytes, gathered from 9 independent experiments. The simulated results in D–F were obtained from 1,000 simulations, while in G, this number was reduced to 70 to match the size of the experimental data. Parameters for box plots in G are as described in Fig 2. $p$-values in G (top row) were obtained using a one-way ANOVA to test the influence of the DNA cluster distance to cortex ratio (in/out) over the $t_i$ rotation. $p$-values in G (bottom row) were obtained using a two-sided Mann–Whitney test. Data underlying this figure can be found in S1 Data. $t_i$ rotation, initial time of rotation.

symmetry. This allowed us to test our model and compare how $t_i$ rotations influenced relevant geometric quantities such as the spindle rotation angle $\alpha$, the DNA cluster distance to cortex, and the depth of ingression in real and simulated oocytes (Fig 7E). We found that our model nicely recapitulated the dynamics of symmetry breaking as seen in delayed rotation oocytes (see slow population in top left panel), which maintain their clusters at similar distance from the cortex for a longer period (middle panel) and undergo a deeper ingression of the cytokinetic furrow (bottom panel).

In the light of this model, and our experimental observations (see Fig 4C), a straightforward prediction was that an initial asymmetry in the DNA cluster position should bias the direction of rotation. To verify this hypothesis, we measured the distance to the cortex of each of the 2 DNA clusters (in and out) and examined the distribution of the averaged DNA cluster distance to cortex ratio (in/out) in the first 10 min of recording (Fig 7F). We observed that, in both simulations and experiments, the distributions were biased to the right, indicating that the extruded cluster (out) tended to be closer to the cortex before the onset of rotation (experiments 76.4%; simulations 83.7% right bias). In addition, we also predicted that a higher initial asymmetry should lead to an earlier start of rotation. We therefore plotted the in/out ratio against the rotation initiation times and found that these 2 quantities were negatively correlated in both simulations and experiments (Fig 7G). This confirmed that the delayed rotation oocytes were initially more symmetric.

Finally, this led us to question whether this simple instability mechanism could also recapitulate the right-tailed distribution of the $t_i$ rotation (top panel, Fig 7D). Running a batch of simulations with Gaussian noise, we found that the distribution of $t_i$ rotation was not Gaussian but skewed to the right, as observed experimentally (bottom panel, Fig 7D). To make further sense of this nontrivial observation, we devised a simplified analytic model of symmetry breaking, in which we calculated the time required to reach an arbitrary symmetry breaking point from an initial configuration that slightly deviated from the unstable equilibrium. We could then calculate the distribution of the symmetry breaking time when the initial offset had a Gaussian distribution. We found that the symmetry breaking time had a right-tail distribution, which was a direct consequence of the nonlinear dependency of the rotation time on the initial offset (see Methods and S6 Fig).

Overall, this model suggests that spindle rotation occurs due to the instability of the parallel configuration as furrow ingression progresses further. A minimal set of ingredients, namely the inward contraction and the lateral outward attractions, are sufficient to recapitulate the dynamic and stochastic aspects of spindle rotation.

## Discussion

Overall, this study reveals that the DNA-induced polarity pathway operating in metaphase II is still active after anaphase onset and underlies the symmetry breaking occurring during spindle rotation. What differs in anaphase II is that the polarized cortex splits into 2 smaller-sized domains overlying each of the segregating DNA clusters. Like in metaphase II, the polarized

cortex exerts attractive forces on the clusters, which, coupled to the ingression of the cytokinesis furrow in the central spindle region, leads to an unstable configuration resulting in spindle rotation. Using theoretical modeling, we showed that this instability stems from the geometric incompatibility of maintaining both DNA clusters at the cortex while pushing the central spindle inward upon cytokinetic ingression. Consequently, any initial asymmetry in the system is sufficient to break symmetry and triggers spindle rotation. Our in vivo measurements confirmed these views, as the initial relative distance of the DNA clusters to the cortex before rotation proved to be a good predictive marker of its direction.

In agreement with Wang and colleagues [32], we believe that cytoplasmic streaming and resulting hydrodynamic forces play a pivotal role in spindle rotation. However, from our perspective, the unilateral contraction of the Ran/Cdc42-induced myosin-II ring (see collapsing protrusion in Fig 3) may not be responsible for the flow inversion, described by these authors as essential for spindle rotation. Indeed, our PIV analysis showed that the inversion occurs early in anaphase II, at a time when none of the cortical protrusions have yet collapsed (S1 Fig). This implies that myosin-II still forms a ring in both protrusions and their contraction cannot be held responsible for the flow reversal. Furthermore, Cdc42 inactivated oocytes still undergo reverse cytoplasmic streaming despite the loss of both the F-actin cap and the myosin-II rings (see Fig 6). Therefore, it appears that the contribution of cytokinetic furrow ingression in reverting the flow was previously overlooked [32].

That said, we would like to stress that, in our view, inverted cytokinetic flows are not instrumental to the symmetry breaking per se. Indeed, the directed spindle spin observed in some PLK1-inhibited oocytes suggests that symmetry breaking can occur in the absence of furrow ingression and reverted flows (Fig 2). According to our model, it is rather the cortical attraction forces that generate the unstable configuration resulting in spindle rotation (Fig 7). This presumably involves another type of cytoplasmic flows, which we referred to as the metaphase-like flows (also referred to as Arp2/3 flows in [32]) that emerge downstream of Ran/Cdc42 signaling and cortical F-actin polarization (Figs 5 and 6). These flows are oriented toward the polarized cortex and were previously described to promote spindle and chromosomes off-centering during late metaphase I and metaphase II [14,20]. We showed here that metaphase-like flows persist during anaphase II (Figs 5 and 6), though they tend to be masked by the stronger inverted cytokinetic flows resulting from furrow ingression. Further investigations will be required to fully understand how these antagonistic flows interfere with each other and influence spindle and chromosome positioning during anaphase II. Improving the spatiotemporal resolution of confocal acquisition, as well as using more discrete tracking algorithms could probably help resolving short-ranged metaphase-like flows throughout the second meiotic division.

If the contraction of the Ran/Cdc42-induced myosin-II ring is not responsible for the early inversion of the flow, we can therefore speculate on its function. First, we cannot exclude that the contraction of the ring contributes to cytoplasmic streaming to some extent, as well as spindle rotation in the later stages of anaphase II. It is also possible that ring contraction drives the rapid collapse of the internalizing cortical protrusion, thereby preventing it from interfering with the PB2 extrusion. Finally, and more generally, the polarized myosin-II ring may be a means for the cell to spatially constrain the F-actin cap and thus better control the attractive forces generated by the cortex. In any case, it will be difficult to draw conclusions about the role of the myosin-II ring and its subsequent contraction without identifying a specific way to inhibit this myosin-II subpopulation. Unfortunately, broad inhibitors such as blebbistatin and ML-7, which effectively prevent myosin-II ring assembly in metaphase II oocytes, were also shown to prevent cytokinesis ring contraction [19,30]. On the other hand, we showed that ROCK1 inhibition targets specifically myosin-II in the cytokinesis furrow (S4 Fig). This raises

the idea that another kinase, acting downstream of Ran/Cdc42, might specifically promote the assembly of the polarized myosin-II rings.

To conclude, as it is often the case in living systems [52], the symmetry breaking process underlying spindle rotation arises from the ability of the oocyte to detect and amplify an initial asymmetry. This is also the case in meiosis I, during which the metaphase I spindle builds upon a slight off-centering to determine the direction of its migration [8]. During meiosis II, our results ultimately suggest that the direction of the rotation, and thus which set of chromatids is eventually discarded, is not biologically predetermined but rather the consequence of a stochastic, spontaneous symmetry breaking process.

## Methods

### Oocyte recovery and culture

All animal procedures were conducted in accordance with the European directive for the use and care of laboratory animals (2010/63/EU) and approved by the local animal ethics committee under the French Ministry of Higher Education, Research and Innovation (Project license APAFIS#11761). Female mice of the OF1 strain (8 to 10 wk old; Charles River) were primed by intraperitoneal injection of 5 to 7 units of PMSG (Chronogest, MSD), followed 48 h later by a second injection of 5 IU hCG (Chorulon, MSD, Intervet, France). Metaphase II oocytes were recovered from the oviducts in M2 medium (Sigma, St Quentin-Fallavier, France) supplemented with 3 mg/ml hyaluronidase (Sigma) followed by wash. To induce resumption of meiosis-II, metaphase II oocytes were treated for 7 min with 7% ethanol, as previously described [22]. When indicated, the culture medium was supplemented with the PLK1 inhibitor Bi-2536 (250 nM; Axon Medchem #1129), the F-actin polymerization inhibitor latrunculin-A (2 μM; Enzo #BML-T119–0100), the ROCK1 inhibitor Y-27632 (50 μM; Merck Millipore #688000), the microtubule polymerization inhibitor nocodazole (1 μM; Calbiochem #487928), or the Arp2/3 inhibitor CK-666 (100 μM; Sigma #SML0006). For anaphase II experiments, inhibitors were added to the culture medium immediately after the ethanol activation, with the exception of nocodazole in Fig 5, which was added during the division. For metaphase II experiments in S4E Fig, oocytes were incubated with Y-27632 for 2 h before fixation.

### Plasmids, cRNA preparation, and microinjection

The following plasmids were used: H2B-mCherry in pcDNA3 (Robert Benezra; Addgene plasmid #20972; [53]), pGEMHE-eGFP-MAP4 (Jan Ellenberg; Euroscarf plasmid #P30518; [54]), eGFP-UtrCH in pCS2+ (William Bement; Addgene plasmid #26737; [48]), pRK5-myc-Cdc42-T17N (Gary Bokoch; Addgene plasmid #12973), pEGFP-RhoA Biosensor (Michael Glotzer; Addgene plasmid #68026), and RanT24N in pcDNA3.1 (a gift from Ben Margolis; [55]). All inserts were subcloned into pcDNA3.1, as required. The full-length ECT2 sequence was amplified from mouse oocyte cDNA and cloned into pcDNA3.1 downstream of eGFP to generate the eGFP-ECT2 construct. Because oocytes expressing EGFP-Ect2 were found to be highly deformed following activation, we truncated Ect2 of its catalytic half (Ect2N; aa 1–452) to generate EGFP-Ect2N, which is sufficient to study localization at the central spindle and regulation by PLK1 [45], while preserving oocyte shape. Polyadenylated cRNAs were synthesized in vitro from linearized plasmids, using the mMessage mMachine T7 kit and Poly(A) Tailing kit (Ambion, Thermo Fisher Scientific, Waltham, MA, USA), purified with RNeasy purification kit (Qiagen, Hilden, Germany), and stored at −80˚C. Using the electrical-assisted microinjection technique, which allows for a high rate of oocyte survival [56], metaphase II oocytes were injected with approximately 5pl cRNA and cultured for approximately 3 h to allow for protein expression.

## Immunofluorescence

For actomyosin staining, the oocytes were fixed for 25 min at room temperature (RT) with paraformaldehyde 3% in PBS, freshly prepared from a 16% methanol-free paraformaldehyde solution (Electron Microscopy Sciences, Hatfield, PA, USA), and adjusted to pH 7.5 with NaOH. For RhoA, Anillin and ROCK1 staining, oocytes were fixed for 10 min at 4˚C with pre-cooled 10% trichloroacetic acid (Sigma). Next, fixed oocytes were permeabilized with 0.1% Triton X-100 (Sigma) in PBS for 15 min at RT, blocked with 3% BSA (Fraction V; Sigma) in PBS for 2 h at RT, and incubated overnight at 4˚C with primary antibodies in PBS-BSA 3%. On the next day, oocytes were washed in PBS-BSA 3% and incubated with secondary antibodies diluted 1:1,000 in PBS-BSA 1%, for 45 min at 37˚C. The following primary antibodies were used: phospho-myosin light chain 2 (Ser19), pMLC2 (1:200, Rabbit polyclonal, Cell Signaling Technology #3671), RhoA (1:100, mouse monoclonal; Santa Cruz sc-418), Anillin (1:100, rabbit polyclonal; Santa Cruz sc-67327), ROCK1 (1/100, goat polyclonal; Santa Cruz sc-6056), and Rac GAP1 (MgcRacGAP; 1:100, mouse monoclonal; Santa Cruz sc-271110). Secondary antibodies were Alexa Fluor 488-conjugated donkey anti-goat, donkey anti-rabbit, and goat anti-mouse (Invitrogen, Carlsbad, CA, USA). F-actin were stained with Alexa Fluor 568-phalloidin (1:50, Life Technologies). Chromatin was stained with To-Pro-3 (1:200, Invitrogen T3605).

## Live microscopy

Oocytes were placed on glass-bottom dishes (MatTek, Ashland, MA) in a small drop of medium covered with mineral oil. For live imaging in the presence of inhibitors, oocytes were placed on polylysine-coated glass-bottom dishes (MatTek, Ashland, MA) in a total volume of 3 ml, without oil. Oocytes were imaged with a Leica SP5 or SP8 confocal microscope, using a 63× or 20× oil immersion objective. For 63× live imaging, a single confocal Z-plane was acquired every 1 min. For 20× live imaging, 20 confocal Z-planes separated by 3.2 μm were acquired every 2 min. In both cases, the temperature was maintained at 37˚C using a stage top incubator (INUBG2E-GSI, Tokai Hit, Shizuoka-ken, Japan) fitted on the microscope stage.

## Image processing and data analysis

**Software.** All image processing and data analyses were performed using ImageJ 1.52t or Matlab 2015a either separately or together using the MIJ plugin (D. Sage, D. Prodanov, C. Ortiz, and J. Y. Tivenez, retrieved from http://bigwww.epfl.ch/sage/soft/mij). The graphics were produced using OriginPro 9.0 and exported to Adobe Illustrator CS6 for final figure layout.

**Live 3D segmentation and tracking procedure.** To monitor the position of the chromatid clusters within the oocyte, we developed an automatic 3D segmentation and tracking procedure coded in Matlab and ImageJ. This method, based on gray levels thresholding, enabled us to extract the DNA clusters and the cell outlines in H2B-mCherry–injected oocytes. Briefly, we filtered the images using Gaussian blurs to smoothen the edges and applied adaptive thresholding to account for the overall brightness reduction in the deeper part of the Z-stacks. We then used different threshold parameters to separate the DNA clusters from the cytoplasmic signal and finally tracked the DNA clusters over time using a proximity criterion. This method allowed us to extract the following parameters: (1) The distance $d$ between the DNA clusters, which was obtained by measuring the 3D Euclidean distance between the 2 DNA clusters. We used the variation of this distance to register in time the oocyte population. To do so, we manually selected early time points for which the distance increases linearly. We next fitted these first few time points and defined the intercept of the linear fit and the time axis as t = 0

(see Fig 1D). (2) The spindle rotation angle $\alpha$, which was obtained by measuring in 3D the angle formed by the internalized cluster of chromatids ($C_{In}$), the spindle midpoint (equidistant from the 2 clusters), and the oocyte centroid (see Fig 1E). (3) The spindle spin angle $\beta$, which was obtained by measuring in 3D the angle formed by the spindle midpoint at the beginning of rotation, the spindle midpoint at the end of rotation, and the oocyte centroid (see S3C Fig). (4) The DNA cluster distance to cortex, which was obtained by averaging the 3D Euclidean distance between the DNA clusters and their 1% closest cortical pixels, resulting from the cytoplasmic outline segmentation. (5) The spindle midpoint distance to the oocyte centroid, which was obtained by measuring the 3D Euclidean distance between the spindle midpoint and the oocyte centroid. In S5E and S5F Fig, we linearly fitted temporal variation of this distance (within a 10- to 50-min analysis window) to extract slopes that were interpreted as the spindle relocalization speed toward the oocyte centroid. (6) The $t_i$ rotation, which was defined as the time when the spindle reaches 5% of its total fitted rotation. To fit the rotation, we manually selected the time points corresponding to the actual rotation and fitted the results using a 5-parameter logistic model implemented in Matlab (G. Cardillo, retrieved from https://github.com/dnafinder/logistic5).

**Cortical F-actin and myosin-II levels and distance to DNA clusters.** During this study, we repeatedly assessed the cortical F-actin and myosin-II (pMLC2) levels and eventually correlated them to the distance separating the cortex and the DNA clusters (see Figs 3C, 3D, 4B, and 4C). To do so, we first segmented the oocyte, using the procedure described above, and defined a 10- to 20-pixel-wide line selection following the cell outline. We next generated Euclidean distance maps (EDM commands in Matlab and ImageJ) centered on the centroid of the DNA clusters. These maps use grayscales to encode the distances between a previously defined point (for instance, DNA cluster centroid) and the rest of the image pixels. Finally, we extracted the values along the line selections for the F-actin, myosin-II, and EDM channels. In Fig 3B (cortical fluorescence intensity profiles), we simply plotted the cortical F-actin and myosin-II levels as a function of the linear cortical distance to the cortical point situated at equal distance from the 2 DNA clusters. In Figs 3D and S5B, we averaged profiles from different oocytes showing the cortical F-actin and myosin-II levels as a function of the Euclidean distance from the cortex to the DNA clusters. In Fig 4B, we generated 2D color-coded maps of the polarized domain showing the variation over time of the cortical F-actin levels or the cortical distance to the DNA clusters. In Fig 4C, we recapitulated the color-coded maps in 1D by averaging results for the 25% (cortical F-actin levels) or the 1% (cortical distance to the DNA clusters) closest cortical pixels.

**Particle image velocimetry (PIV).** To measure the flow of cytoplasmic materials in oocytes, we devised a Matlab routine built on the PIVlab implementation of PIV [57]. We eliminated the vectors outside the oocyte using a mask and smoothed the vector field in space and time with custom filters based on running averages. In Figs 1C, 2H, 5E, 6F, and S1E Fig, we used an interrogation window of $80 \times 80$ pixels for the first pass and $40 \times 40$ pixels for the second pass. We smoothed the vector field using a $3 \times 3$ vectors kernel in space and a 9–time point kernel in time. In S1A Fig, we used an interrogation window of $160 \times 160$ pixels for the first pass and $80 \times 80$ pixels for the second pass. We smoothed the vector field using a $5 \times 5$ kernel in space and a 13–time point kernel in time. Finally, for display purposes, we color coded the vector norm using the "quiverc" function (B. Dano, retrieved from https://www.mathworks.com/matlabcentral/fileexchange/3225-quiverc). To average the vector field in Figs 2H, 6F, and S1E Fig, we oriented all oocytes in the same direction before running the PIV analysis and averaged vectors for all interrogation windows. To extract the flow velocity curves in S1D, S1F and S5C Figs, we restricted the PIV analysis to the central part of oocytes (see PIV ROI in S1B Fig) and extracted components of velocity vectors along the axis defined by the

polarized domain and the opposite side of the cell. By doing so, positive values indicate a flow oriented toward the oocyte center (inverted flow), while negative values indicate a flow oriented toward the polarized domain (metaphase-like flow). To extract the max PIV velocity in S5D and S5G Fig, we averaged the 10% maximum velocities recorded during an analysis window ranging from 10 to 50 min after activation.

## Numerical modeling

In the model, we considered a spindle of fixed length in a circular oocyte. DNA clusters are localized at the ends of the spindle. As described in the main text, the spindle is essentially submitted to 2 opposing forces: the attraction of each cluster to the locally polarized cortex, and the cytokinetic ingression pushing the central spindle inward. A schematic of the model geometry is depicted Fig 7A.

**Attraction to the cortex.** We modeled the attraction of DNA clusters to the cortex through an effective potential. For simplicity, we used a soft core–like potential $V$ of the form (Fig 7B):

$$V(r) = A(r - r_a + \lambda)e^{-r/\lambda}$$

where $r$ is the distance to the attraction zone of the cortex, and $\lambda$ is the typical decay length of the potential. $r_a$ is the value where $V(r)$ is minimum and typically corresponds to the radius of DNA clusters: $V$ is minimum when the cluster is apposed to the attraction zone.

The force $\overrightarrow{f}_V$ exerted by each attraction zone on the corresponding cluster is simply the gradient of $V(r)$:

$$\overrightarrow{f}_V = -\frac{\partial V}{\partial r}\overrightarrow{u}_r$$

**Central ingression.** The cytokinetic ingression acts as a pushing force on the spindle midpoint (corresponding to the central spindle). For simplicity, we assumed in the model that the ingression velocity is constant. Numerically, we simply prescribed the velocity of the central spindle, which moves with constant velocity $\overrightarrow{u}_{cyto}$ perpendicular to the spindle. Importantly, without this prescribed ingression, the clusters remain at the cortex and nothing happens.

**Spindle rigidity.** We allowed the spindle to slightly bend, as observed in experiments. Bending of the spindle yields an elastic restoring force on clusters, which is proportional to the deflection $\overrightarrow{w}$ of the spindle and to the spindle stiffness $k$, so that the elastic force is $\overrightarrow{f}_E = k\overrightarrow{w}$. The deflection is equal to the distance between the midpoint of the straight line joining the 2 clusters and the midpoint of the spindle. Note that $\overrightarrow{f}_E$ is perpendicular to the straight line joining the clusters. It is because of the rigidity of the spindle (accounted for by the elastic restoring force) that the clusters follow the movement of the central spindle prescribed by $\overrightarrow{u}_{cyto}$.

**Time evolution.** We obtained the movement of each cluster using overdamped dynamics:

$$\overrightarrow{v} = \frac{1}{\alpha}\left(\overrightarrow{f}_V + \overrightarrow{f}_E\right)$$

where $\overrightarrow{v}$ is the velocity of a given cluster, and $\alpha$ the damping coefficient. To compute the new position of each cluster at each time step, we integrated this equation numerically using a simple Euler method.

**Noise.** This problem has an unstable solution, in which the spindle remains parallel to the cortex while being pushed by the ingression. In that case, both clusters eventually escape their attraction potential. This unstable solution cannot be observed in vivo, as any noise, asymmetry, or perturbation will lead to symmetry breaking. The unstable solution can be observed numerically if no noise whatsoever is introduced (Fig 6C), as the simulation is then perfectly symmetric. However, as soon as a very small noise is introduced, the system eventually breaks its left–right symmetry, with one cluster escaping its attraction potential while the other returns to its attraction zone (Fig 7C).

There are many possible sources of noise in the biological system, which may include (but are not limited to) an asymmetry of the initial configuration, an asymmetry of the decay length $\lambda$, an asymmetry of the attraction force, thermal or active fluctuations of the clusters' position, an imperfect centering of the central spindle or of the ingression, etc. Here, we only considered noise on the dynamics of clusters position.

Upon integration of the equations of motion, we added a normally distributed random noise on each cluster's position at each time step, with standard deviation $\sigma$. This is sufficient to break the symmetry of the system even when $\sigma$ is very small compared to the system size. Note that this also introduces a normally distributed initial asymmetry, which inevitably sets a bias for the direction of rotation (Fig 7F).

**Parameters values.** In simulations, we essentially used the same set of parameters. The only changes concerned the noise introduced to make better sense of the symmetry-breaking mechanism and of the variability observed experimentally. Note that distances are in units of $R_{ovo}$, the radius of the oocyte (we set $R_{ovo} = 1$). The timescale is arbitrary, as we set the damping coefficient to 1.

Parameters used throughout the study are as follow:

$\lambda = 1 \times 10^{-1}$

$r_a = \frac{4}{3} \times 10^{-1}$

$\alpha = 1$

$A = 1.5 \times 10^2$

$k = 2 \times 10^2$

$u_{cyto} = 1/3$

Noise:

Fig 7C (simulations, top panel): $\sigma = 0$

Fig 7C (simulations, other panels): $\sigma = 1 \times 10^{-3}$

Fig 7D–7F (angle and distance dynamic): $\sigma = 1 \times 10^{-3}$

Fig 7G (int/out histogram, in/out and rotation time): $\sigma = 1 \times 10^{-2}$

**Simplified analytical model of the "symmetry-breaking time" distribution.** To make better sense of the long-tailed distribution of the symmetry breaking time, observed both in vivo and in silico, we designed a simplified model of the symmetry breaking process. Let us consider the dynamics of an object on a locally quadratic bump of potential $V$:

$$V(r) = -\frac{1}{2} k r^2$$

where $r$ is the position of the object, and $a$ is a constant. The overdamped dynamics of the object is simply given by:

$$\dot{r} = -\frac{1}{\alpha} \frac{\partial V}{\partial r} = \frac{1}{\tau} r$$

where $\alpha$ is the damping coefficient, and $\tau = \frac{\alpha}{k}$ is the relaxation timescale. The position of the

object at time $t$ is thus given by:

$$r(t) = r_0 e^{t/\tau}$$

If $r_0 = r(t = 0) = 0$, the solution is $r(t) = 0$ (the object remains on the maximum of $V(r)$). Clearly, this is an unstable solution, as any infinitesimal deviation from $r_0 = 0$ will increase exponentially over time, with the object moving away from the maximum of $V(r)$. Let us now consider the time $t_c$ (the "symmetry-breaking" time) needed for the object to reach an arbitrary distance $r_c$ considered the symmetry-breaking distance, starting from $r_0 \neq 0$. Taking the log of the equation above, we have:

$$t_c = \tau \ln(r_c/r_0)$$

Let us now assume that $r_0$ has a Gaussian probability density function (with standard deviation $\sigma$), as in our batch of oocytes simulations: $p(r_0) = \frac{1}{\sigma\sqrt{2\pi}} e^{-\frac{1}{2}(r_0/\sigma)^2}$. Using the expression of $t_c$ above and the probability density function of $r_0$, one can directly calculate the probability density function of the symmetry-breaking time $t_c$, which reads:

$$p(t_c) = \frac{2r_c}{\sigma\tau\sqrt{2\pi}} e^{-t_c/\tau} e^{-\frac{1}{2}(r_c e^{-t_c/\tau}/\sigma)^2}$$

With the analytical expression above, we directly recover the long-tailed distribution $p(t_c)$ observed in our simulations and in vivo, starting from a Gaussian noise (S6 Fig, top and middle panels). Note that computing $p(t_c)$ numerically from a normal distribution of $r_0$ yielded the same result. The long tail of $p(t_c)$ can be viewed as a direct consequence of the nonlinear decay of $t_c(r_0)$ close to the unstable solution. Small values of $r_0$ lead to a long symmetry-breaking time (S6 Fig, bottom panel). For plots shown in S6 Fig, we used $\sigma = 1$, $\tau = 1$, and $r_c = 10$.

## *p*-values and reproducibility

Data points from different oocytes were pooled to estimate the mean and standard deviations. The quantifications were carried out on a minimum number of 10 oocytes from a minimum number of 2 independent experiments. Statistical significance was tested using Mann–Whitney tests, assuming nonnormal distributions and equal variance. For the data in Fig 7G and S5G Fig (top row), we performed a one-way ANOVA to test the influence of the Cortex/DNA cluster distance ratio over the $t_i$ rotation. No statistical method was used to predetermine sample sizes. The experiments were not randomized, and the investigators were not blinded to allocation during experiments and outcome assessment.

## Supporting information

**S1 Fig. Cytoplasmic streaming during spindle rotation.** (A) PIV measurements showing cytoplasmic streaming during the course of anaphase (S4 Movie). Still images show the reverse flows induced by the cytokinetic furrow ingression (t = 15 min), the collapsing protrusion (t = 40 min), and the closure of the cytokinesis ring (t = 55 min). (B) Diagram illustrating the method used for 1D flow velocity measurements (see Methods). The dotted frame indicates the ROI for PIV measurements. Positive values indicate a flow oriented toward the oocyte center (reverse flow), while negative values indicate a flow oriented toward the polarized domain (metaphase-like flow). (C) Diagram illustrating the centroid distance to the cortical invagination (red dot) or the collapsing protrusion (blue dot). (D) Graph showing the variation over time of flow the velocity (black curve) and the centroid distance to cortical invagination (red curve) or collapsing protrusion (blue curve). (E) PIV vector fields showing the averaged

cytoplasmic streaming pattern in metaphase II and anaphase II oocytes (see Methods). (F) Averaged variation over time ± SD of flow velocity in metaphase II (light gray curve) and anaphase II oocytes (dark gray curve). Measurements in E and F were performed on 11 metaphase II and 32 anaphase II oocytes, respectively, gathered from 2 and 5 independent experiments. Data underlying this figure can be found in S1 Data. Scale bars = 10 μm. PB2, second polar body; PIV, particle image velocimetry; ROI, region of interest.
(TIF)

**S2 Fig. Automated 3D segmentation and tracking procedure.** (A) Automated 3D segmentation and tracking procedure to monitor the position of the DNA clusters within the oocyte (see Methods and S5 Movie for further details). (B) Diagram showing the method used to quantify DNA cluster distance $d$ and the spindle rotation angle $\alpha$. Scale bars = 10 μm.
(TIF)

**S3 Fig. Recruitment of canonical cytokinesis factors during anaphase II.** (A) Fixed RhoA, anillin, ROCK1 (immunostaining), DNA (To-Pro-3), and eGFP-ECT2N imaging of metaphase II (insets) and anaphase II oocytes (main images). Fluorescence patterns shown in A were consistently observed in a minimum of 14 oocytes gathered from a minimum of 2 independent experiments. (B) Fixed eGFP-Ect2-N and MgcRacGAP (immunostaining) imaging on control and Bi-2536–treated oocytes. The PLK1 inhibition prevented eGFP-ECT2N recruitment at the central spindle (top panels) while, in contrast, the recruitment of MgcRacGAP (immunestaining) was preserved (bottom panels). Note the enhanced MgcRacGAP signal in Bi-2536–treated oocytes, which may result from epitope unmasking, due to the release of bound ECT2. Fluorescence patterns shown in B were consistently observed in 23 oocytes (eGFP-ECT2N) and in 15 oocytes (MgcRacGAP) both gathered from 3 independent experiments. (C) Box plots showing the maximum variation of the spin angle $\beta$, as defined in top diagram and Methods, in control and Bi-2536–treated oocytes. Measurements in C were performed on 70 control oocytes gathered from 9 independent experiments and 35 Bi-2536 oocytes gathered from 6 independent experiments. Parameters for box plots and statistics in C are as described in Fig 2. $p$-value in C was obtained using a two-sided Mann–Whitney test. Data underlying this figure can be found in S1 Data. Scale bars = 10 μm. PLK1, Polo-like kinase 1; ROCK1, Rho-associated protein kinase 1.
(TIF)

**S4 Fig. Anaphase II involves 2 differentially regulated cortical actomyosin subpopulations.** (A) Fixed F-actin (phalloidin staining), eGFP-AHPH (RhoA sensor), and DNA (To-Pro-3) imaging of control and latrunculin A–treated anaphase II oocytes. The white arrows show the F-actin and activated RhoA patch facing the central spindle. Note the persistence of RhoA activation despite the lack of F-actin accumulation upon latrunculin A treatment. Fluorescence patterns shown in A were consistently observed in a minimum of 8 oocytes gathered from a minimum of 3 independent experiments. (B) Cortical fluorescence intensity profiles recapitulating F-actin and eGFP-AHPH polarization patterns observed in A. (C) Fixed F-actin (phalloidin staining), myosin-II (pMLC2 immunostaining), and DNA (To-Pro-3) imaging of control, Bi-2536–treated (PLK1 inhibitor), and Y-27632–treated (ROCK1 inhibitor) anaphase II oocytes. The white arrows show the F-actin and myosin-II patch facing the central spindle. (D) Cortical fluorescence intensity profiles recapitulating F-actin and myosin-II polarization patterns observed in C. Note the disappearance of the central spindle-facing actomyosin patch upon PLK1 and ROCK1 inhibition. (E) Fixed F-actin (phalloidin staining) and myosin-II (pMLC2 immunostaining) imaging of control or Y-27632–treated metaphase II oocytes. (F) Fixed myosin-II (pMLC2 immunostaining) imaging of control or Y-27632–treated anaphase

II oocytes. Images show maximum intensity z-projections of multiple confocal planes. Note the persistence of the F-actin cap and the myosin-II ring upon PLK1 and ROCK1 inhibition in C–F. Fluorescence patterns shown in C–F were consistently observed in a minimum of 18 oocytes gathered from a minimum of 3 independent experiments. Data underlying this figure can be found in S1 Data. Scale bars = 10 μm. F-actin, actin filament; PLK1, Polo-like kinase 1; pMLC2, phosphorylated myosin light chain 2; ROCK1, Rho-associated protein kinase 1.
(TIF)

**S5 Fig. Cdc42 promotes cortical attraction forces through F-actin polarization.** (A) Fixed F-actin (phalloidin staining), myosin-II (pMLC2 immunostaining), and DNA (To-Pro-3) imaging of metaphase II–arrested oocytes expressing dominant negative forms of the Ran and Cdc42 GTPase (respectively, RanT24N and Cdc42T17N). (B) Averaged ± SD cortical F-actin (phalloidin staining) and myosin-II (pMLC2 immunostaining) fluorescence intensities as a function of the cortex distance to DNA clusters. The fluorescence intensities are normalized to the mean of each profile. Measurements were performed on 10 controls, 11 RanT24N, and 15 Cdc42T17N metaphase II oocytes, gathered from 2 independent experiments. (C) Graph showing the averaged variation over time ± SD of flow velocity (measured as described in S1 Fig) in control (light gray) and Cdc42T17N oocytes (blue curves). (D) Box plots showing the distribution of maximum flow velocities (average of the 10% highest values) recorded within the 10- to 50-min analysis window (see C and Method). (E) Graph showing the averaged variation over time ± SD of the distance from the oocyte centroid and the DNA cluster midpoint in control (light gray) and Cdc42T17N oocytes (blue curves). The dotted lines represent linear fits performed within the 10- to 50-min analysis window. (F) Box plots showing the distribution of slopes extracted from the linear fits described in E. The slope is interpreted as the spindle relocation speed toward the oocyte centroid (see Method). (G) Scatter plot of spindle speed toward the oocyte centroid versus the maximum flow velocity. Measurements in C–G were performed on 32 control and 29 Cdc42T17N oocytes, respectively, gathered from 8 and 3 independent experiments. Parameters for box plots in D and F are as described in Fig 2. *p*-values (against controls) in D and F were obtained using a two-sided Mann–Whitney test. *p*-values in G was obtained using one-way ANOVA to test the influence of maximum flow velocities over the spindle relocation speed toward the centroid. Data underlying this figure can be found in S1 Data. Scale bars = 10 μm. F-actin, actin filament; PIV, particle image velocimetry; pMLC2, phosphorylated myosin light chain 2; sPB, small polar body.
(TIF)

**S6 Fig. Simplified analytic model of symmetry breaking.** (A) Gaussian distribution of initial deviation from the unstable configuration. The red solid line shows the analytical distribution. (B) Distribution of the resulting symmetry breaking time, required to reach an arbitrary distance from the unstable configuration. The red solid line shows the analytical distribution. (C) Symmetry breaking time vs. initial offset. The red solid line shows the analytical expression. Data underlying this figure can be found in S1 Data.
(TIF)

**S1 Data. Numerical data underlying main and supplementary figures.** Figs 1D, 1E, 1G, 1H, 2D, 2F, 2G, 3B, >3D, 4B, 4C, 5B, 5C, 6C, 6D, 7B, 7D, 7E, 7F, 7G, and S1D, S1F, S3C, S4B, S4D, S5B, S5C, S5D, S6A, S6B and S6C Figs are presented in separate Excel sheets that are combined into a single Excel file.
(XLSX)

**S1 Movie. Anaphase II spindle rotation (brightfield).** Live imaging of an activated oocyte undergoing spindle rotation. The oocyte was injected with the H2B-mCherry DNA marker.

Images were acquired every 1 min with a 63× objective and represent a single confocal Z-plane. Scale bar = 10 μm.
(AVI)

**S2 Movie. Anaphase II spindle rotation (MTs).** Live imaging of an activated oocyte undergoing spindle rotation. The oocyte was injected with a combination of the H2B-mCherry DNA marker and the eGFP-MAP4 MT marker. Images were acquired every 2 min with a 20× objective and represent a maximum projection of 20 confocal Z-planes. Scale bar = 10 μm. MT, microtubule.
(AVI)

**S3 Movie. Cytoplasmic streaming pattern in metaphase II and anaphase II oocytes (PIV).** Live imaging comparing cytoplasmic streaming in metaphase II and anaphase II oocytes. Note the flow reversal in anaphase II. Oocytes were injected with the H2B-mCherry DNA marker. Images were acquired every 2 min with a 20× (metaphase II) or 63× (anaphase II) objective and represent a single confocal Z-plane. The PIV vector field color code indicates the speed of tracked particles. The top-right arrows show size and color of a 0.3-μm.min-1 vector. Scale bar = 10 μm. PIV, particle image velocimetry.
(AVI)

**S4 Movie. Cytoplasmic streaming pattern in anaphase II oocytes (PIV, high temporal resolution).** Live imaging revealing cytoplasmic streaming in an activated oocyte. The oocyte was injected with the H2B-mCherry DNA marker. Images were acquired every 1 min with 63× objective and represent a single confocal Z-plane. The PIV vector field color code indicates the speed of tracked particles. The top-right arrows show size and color of a 0.35-μm.min-1 vector. Scale bar = 10 μm. PIV, particle image velocimetry.
(AVI)

**S5 Movie. Automated 3D segmentation and tracking procedure.** Automated 3D segmentation and tracking procedure to monitor the position of DNA clusters within the oocyte. The top panel shows tracked DNA clusters from 3 differentially oriented oocytes, while the bottom panel shows a 3D reconstruction of the same oocyte volumes. Images were acquired every 2 min with a 20× objective and represent a maximum projection of 20 confocal Z-planes.
(AVI)

**S6 Movie. Stochastic triggering of spindle rotation.** Live imaging of 2 activated oocytes showing a clear difference in their rotation initiation time. Oocytes were injected with the H2B-mCherry DNA marker. Images were acquired every 2 min with a 20× objective and represent a maximum projection of 20 confocal Z-planes. Scale bar = 10 μm.
(AVI)

**S7 Movie. Central spindle/RhoA pathway inhibition (Bi-2536).** Live imaging of control and Bi-2536–treated (PLK1 inhibitor) activated oocytes. Oocytes were injected with the H2B-mCherry DNA marker. Among the Bi-2536–treated oocytes, about one-third ($n$ = 10/35 approximately 29%) of the spindles engaged in a spin around the cell periphery (see right oocytes), while the others ($n$ = 25/35 approximately 71%) simply remained parallel to the cortex (see middle oocyte). Images were acquired every 2 min with a 20× objective and represent a maximum projection of 20 confocal Z-planes (H2B-mCherry) or a single confocal Z-plane (brightfield). Scale bar = 10 μm. PLK1, Polo-like kinase 1.
(AVI)

**S8 Movie. Cortical F-actin reorganization during anaphase II.** Live imaging of an activated oocyte undergoing spindle rotation. The oocyte was injected with a combination of the H2B-mCherry DNA marker and the eGFP-UtrCH F-actin marker. Images were acquired every 2 min with a 20× objective and represent a maximum projection of 20 confocal Z-planes. Scale bar = 10 μm. F-actin, actin filament.
(AVI)

**S9 Movie. Nocodazole-induced DNA cluster detachment reveals cortical attraction forces.** Live imaging of an activated oocyte treated with low dose of nocodazole. This treatment occasionally resulted in DNA cluster detachment from the spindle and revealed that DNA clusters undergo attractive forces from the polarized cortex. The manually tracked cytoplasmic particles (see white circles) and the PIV measurements showed that metaphase-like flows, oriented toward the polarized cortex, persist during anaphase II. The oocyte was injected with a combination of the H2B-mCherry DNA marker and the eGFP-MAP4 microtubule marker. Images were acquired every 2 min with a 20× objective and represent a maximum projection of 20 confocal Z-planes (H2B-mCherry + eGFP-MAP4) or a single confocal Z-plane (brightfield). The PIV vector field color code indicates the speed of tracked particles. The top-right arrows show size and color of a 0.4-μm.min-1 vector. Scale bar = 10 μm. PIV, particle image velocimetry.
(AVI)

**S10 Movie. Cortical actomyosin polarization inhibition (Cdc42T17N).** Live imaging of activated oocytes injected with the H2B-mCherry DNA marker alone (control, left panel) or in combination with Cdc42T17N. Among the Cdc42T17N-injected oocytes, about one-third ($n$ = 8/27 approximately 30%) of the spindle relocated to the center of the cell (see right oocyte), while the others ($n$ = 19/27 approximately 70%) extruded a PB2, though smaller in size than in controls (see middle oocyte). Note also that, as control, Cdc42T17 oocytes undergo reverse cytoplasmic streaming. Images were acquired every 2 min with a 20× objective and represent a maximum projection of 20 confocal Z-planes (H2B-mCherry) or a single confocal Z-plane (brightfield). Scale bar = 10 μm. PB2, second polar body.
(AVI)

**S11 Movie. Cortical actomyosin and cytokinetic ingression inhibition (Cdc42T17N + Y-27632).** Live imaging of activated oocytes injected with the H2B-mCherry DNA marker alone (control, left panel) or in combination with Cdc42T17N. The culture medium was supplemented with the ROCK1 inhibitor, Y-27632, to prevent the cytokinetic ingression. Note that both the reversion of cytoplasmic streaming and the spindle relocation were prevented in this inhibitory condition. Images were acquired every 2 min with a 20× objective and represent a maximum projection of 20 confocal Z-planes (H2B-mCherry) or a single confocal Z-plane (brightfield). Scale bar = 10 μm. ROCK1, Rho-associated protein kinase 1.
(AVI)

**S12 Movie. Antagonistic forces yield symmetry breaking and spindle rotation.** Time-lapse sequences of typical simulations with (left) or without (middle and right) Gaussian noise added to the DNA clusters position.
(AVI)

## Acknowledgments

We are grateful to the staff of the ARCHE-Biosit animal facility and MRIC-Biosit microscopy facility for technical assistance and expert advice.

## Author Contributions

**Conceptualization:** Benoit Dehapiot, Raphaël Clément, Sébastien Huet, Guillaume Halet.

**Data curation:** Benoit Dehapiot.

**Formal analysis:** Benoit Dehapiot, Raphaël Clément.

**Funding acquisition:** Guillaume Halet.

**Investigation:** Benoit Dehapiot, Anne Bourdais, Virginie Carrière.

**Methodology:** Benoit Dehapiot.

**Project administration:** Guillaume Halet.

**Resources:** Benoit Dehapiot, Anne Bourdais.

**Software:** Benoit Dehapiot, Raphaël Clément.

**Supervision:** Sébastien Huet, Guillaume Halet.

**Validation:** Guillaume Halet.

**Visualization:** Benoit Dehapiot.

**Writing – original draft:** Benoit Dehapiot.

**Writing – review & editing:** Benoit Dehapiot, Raphaël Clément, Guillaume Halet.

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
