## [Editor Report · Decision Letter 0]

19 Oct 2020

Dear Dr Dehapiot, 

Thank you for submitting your manuscript entitled "RhoA- and Ran-induced antagonistic forces underlie symmetry breaking and spindle rotation in mouse oocytes." for consideration as a Research Article by PLOS Biology.

Your manuscript has now been evaluated by the PLOS Biology editorial staff as well as by an academic editor with relevant expertise and I am writing to let you know that we would like to send your submission out for external peer review. Please note, however, that the outcome of our discussion of your manuscript is that we have some reservations as to the overall strength of novel biological insight offered by your data. We would need to be persuaded by the reviewers that the paper has the potential after revision to offer the significant strength of advance that we require for publication in order to pursue it further for PLOS Biology.

Before we can send your manuscript to reviewers, we need you to complete your submission by providing the metadata that is required for full assessment. To this end, please login to Editorial Manager where you will find the paper in the 'Submissions Needing Revisions' folder on your homepage. Please click 'Revise Submission' from the Action Links and complete all additional questions in the submission questionnaire.

Please re-submit your manuscript within two working days, i.e. by Oct 21 2020 11:59PM.

Kind regards,

Ines

--

Ines Alvarez-Garcia, PhD,

Senior Editor

PLOS Biology

---

## [Decision Letter · Decision Letter 1]

21 Dec 2020

Dear Dr Dehapiot,

Thank you very much for submitting your manuscript entitled "RhoA- and Ran-induced antagonistic forces underlie symmetry breaking and spindle rotation in mouse oocytes" for consideration as a Research Article at PLOS Biology. Thank you also for your patience as we completed our editorial process, and please accept my apologies for the delay in providing you with our decision. Your manuscript has been evaluated by the PLOS Biology editors, an Academic Editor with relevant expertise, and by three independent reviewers.

You will see that the reviewers think your study is interesting, thorough and well performed, however they also think it requires more direct evidence to support the existence of the outward forces you propose as a mechanism. They all suggest several experiments you should perform to strengthen your conclusions and ask you to clarify some points.

In light of the reviews (attached below), we will not be able to accept the current version of the manuscript, but we would welcome re-submission of a much-revised version that takes into account the reviewers' comments. We cannot make any decision about publication until we have seen the revised manuscript and your response to the reviewers' comments. Your revised manuscript is also likely to be sent for further evaluation by the reviewers.

We expect to receive your revised manuscript within 3 months. 

**IMPORTANT - SUBMITTING YOUR REVISION**

3. Resubmission Checklist

a) *Published Peer Review*

b) *PLOS Data Policy*

Please make sure you mention in the corresponding figure legends where the data can be found and ensure that your Data Statement in the submission system accurately describes where your data can be found.

d) *Blurb*

Please also provide a blurb which (if accepted) will be included in our weekly and monthly Electronic Table of Contents, sent out to readers of PLOS Biology, and may be used to promote your article in social media. The blurb should be about 30-40 words long and is subject to editorial changes. It should, without exaggeration, entice people to read your manuscript. It should not be redundant with the title and should not contain acronyms or abbreviations. For examples, view our author guidelines: https://journals.plos.org/plosbiology/s/revising-your-manuscript#loc-blurb

Sincerely,

Ines

--

Ines Alvarez-Garcia, PhD,

Senior Editor,

PLOS Biology

Reviewers’ comments

Rev. 1:

Dehapiot et al. analyze here the process of spindle rotation necessary for the extrusion of the second polar body in mouse oocytes. They provide convincing evidence that after parthenogenetic activation, the central spindle initiates furrow ingression via the classical Rho A activation pathway locally promoting cytokinetic myosin II ring formation. This initial event triggers a series of cortical reorganizations, such as actomyosin polarization above the mass of chromosomes that wins the tug of war between the two separated masses at opposite spindle poles, that will further amplify the breaking of symmetry. They also suggest that the polarized cortex exerts forces on the chromatid clusters. Eventually, they propose a very simple model with two antagonistic forces, one exerted by furrow ingression and cortical attraction, that recapitulates some features of spindle rotation.

In the whole I find the iconography presented in this manuscript to be awesome, that most of the experiments are sound and well executed and hence convincing. I would have just two minor comments.

1/ While most of the experimental evidence is convincing, the data on cortical attraction presented in Figure 5 is to me the weakest part. What exactly is the nature of this cortical attraction? Can you detect actin filaments promoted by local Arp2/3 activation connecting the mass of chromosomes or the spindle to the cortex? If you locally inhibit Arp2/3, via CK666 for example, do you also observe the spindle going away from the cortex and local disappearance of potential connections between the spindle and the cortex?

2/ Concerning the modeling, it is nice and simple, but maybe authors should clearly sate from the beginning that it is not quantitative and this is why all outcomes from simulations are presented in arbitrary units. Indeed, the intensities of two antagonistic forces applied on the spindle have not been measured, nor the viscosity of the cytoplasm, etc, etc…

Furthermore, one equation on spindle motion seems to be missing in the methods. It is not clear whether u_cyto, the ingression force, is computed on the center of the spindle (pivot point) or on both clusters of chromosomes. How also is the W deflection computed and whether it allows to keep a constant spindle length. The issue of spindle flexibility could be better explained.

Eventually, in the text (lane 292), it would be nice to explain to an audience of cell biologists where the noise comes from (rather than putting it in the methods section, lane 735). Indeed it is most certainly the noise which allows the breaking of symmetry so it is important to explain and discuss it in the main text.

Rev. 2:

The perpendicular orientation of oocyte meiotic spindles at the cortex is thought to be essential for expulsion of the correct number of chromosomes into a polar body to prevent lethal aneuploidy. Thus the problem of meiosis II spindle rotation is highly significant. For mouse, the basic features of a single actin-dependent protrusion over the metaphase chromosomes that splits into two protrusions over the separating sets of anaphase chromosomes, then resolve into a single polar body, were described by Bernard Maro in 1984. This manuscript does not reference that paper. Throughout the 21st century, a number of labs have rediscovered actin-dependent meiotic spindle positioning in mouse oocytes and documented a number of molecules required but have not elucidated the actual mechanisms positioning the spindle. This manuscript uses modern microscopy methods to recapitulate Maro’s work and to demonstrate that several molecules already known to be active at metaphase, continue to be active during anaphase and spindle rotation. The manuscript does not elucidate the mechanism that moves one set of chromosomes toward the cortex during rotation which is the significant outstanding question. Using 3D projections of F-actin and myosin II light chain, the authors nicely show that one contractile ring over the metaphase chromosomes splits into 2 contractile rings, one over each set of anaphase chromosomes, then one protrusion relaxes as one set of chromosomes rotates into the stable protrusion. While very beautifully shown, the advance is somewhat incremental since one protrusion splitting into two protrusions that resolve again into one protrusion was described by Bernard Maro in 1984 (36 years ago). This manuscript addresses a specific question common to all mitotic and meiotic spindle positioning situations. If a force moves both ends of the spindle toward the cortex, how does one end of the spindle pivot ahead of the other rather than becoming stalled with both spindle ends equidistant from the cortex. In mouse meiosis I spindle migration, micromanipulation was previously used to definitively show that the end of the spindle that is fortuitously closer to the cortex leads toward the cortex. In this manuscript, the authors use correlative observation and a mathematical model to suggest that the end of the spindle that is fortuitously closer to the cortex will lead during meiosis II rotation. By adding the assumption that velocity increases with proximity to the cortex, they can recapitulate rotation that requires stochastic noise in the initial parallel orientation. They do not do the micromanipulation experiment to demonstrate cause and effect. The result is thus not surprising because it is the same as the meiosis I result and it is not as rigorously demonstrated as the meiosis I result. The manuscript provides a very nice description of mouse meiosis II spindle rotation but does not provide a significant increase in mechanistic understanding as it is currently written. It is possible that the authors could improve the manuscript by explaining more clearly what is new.

Details:

The mathematical model involves an inward force on the middle of the spindle generated by the ingressing actomyosin furrow (strong premise and molecular mechanism) and an “attractive force” on each set of separating chromosomes (weaker premise and no molecular mechanism). Rong Li’s lab suggested that cytoplasmic flow pushes the metaphase spindle toward the cortex but flow reverse during anaphase leaving no demonstrated model for the “attractive force”. The possibility that conventional dynein and microtubule-dependent pulling is involved, as occurs in C. elegans, appears to be dismissed while 2 publications reported rotation failure after partial microtubule depolymerization in rodent oocytes. This work offers only incremental advances unless they elucidate the mechanism of the attractive force.

Dominant negative cdc42 causes metaphase spindles to move away from the cortex at the same velocity as the cytoplasm flows away from the cortex. This is interpreted to mean that cortical cdc42 generates an attractive force that can resist cytoplasmic streaming. The molecular mechanism of this attractive force is not elucidated making the results highly descriptive rather than mechanistic.

The investigators find that spindle rotation does not occur at a precise time relative to the extent of anaphase chromosome separation. This is an interesting observation supporting their idea that rotation is initiated by stochastic movement of one end of the spindle closer to the cortex.

During meiosis I spindle migration, the cortex proximal spindle pole leads and this has been demonstrated by micromanipulation. The current study does not use micromanipulation and thus only shows a correlation rather than cause and effect.

The authors should address whether ethanol-induced spindle rotation is the same as fertilization-dependent spindle rotation?

Fig. S1 demonstrating reversal of cortical flow at anaphase onset appears to be a single example and the average result from many oocytes is not presented.

Line 104: The direction or nature of the “distinct flows” occurring at the time of protrusion collapse and ring closure are not described and the word “generated” should be replaced with “correlated in time” since no cause and effect relationship is demonstrated.

What are the particles being tracked in the PIV in figure S1 and what is the evidence that velocities represent the same particle that remains in focus for the duration of the velocity measurement?

Line 134: Localization of ECT2, anillin and rho kinase is referenced to Fig. S1 but this data is in Fig. S2. This data needs to be quantified. How many oocytes exhibited the pattern shown? How many did not exhibit the pattern shown?

It would be more of an advance to show co-localization of rhoA with F-actin and myosin II and localization of rhoA in latrunculin treated oocytes so that the reader can see whether rhoA simply co-localizes with F-actin or if there is a cause-effect relationship.

Line 142: The authors should clarify whether the quantitation presented for polar body extrusion also applies to the failure to localize rhoA and the failure in ingression. Quantitation of the number of oocytes exhibiting the flow pattern in Fig. 2G is needed.

The finding that a Plk1 inhibitor prevents rhoA localization, ingression and rotation is interesting. However, Plk1 is involved in many different pathways. The authors should at least show correct localization of MgcRacGAP and localization failure for ECT2 in response to the Plk1 inhibitor if the results are to be related to the pathway diagrammed in Fig. 1A.

Line 157: Do the authors mean “carried by the metaphase II spindle”?

Fig.3A: The third column should be labeled (presumably pMLC2).

Line 176 and Fig S3 need quantitation. How many oocytes exhibited this result and how many did not?

Line 240: “we measured a speed of about 0.3 μm.min-1,” must be replaced with a mean, sem and n.

Throughout the manuscript, the authors refer to the cortex generating “attractive forces on the chromosomes” but the mechanism of these attractive forces is not elucidated in this manuscript.

Rev. 3:

The present study provides new insights into the rotation of meiotic spindle in mouse meiosis - a process that has been largely overlooked in mouse oocytes and its precise mechanisms have remained elusive. By having established a quantitative and systematic approach in live oocytes, the authors provide some evidence that spindle rotation may be directly controlled by membrane invagination during cytokinesis/polar body extrusion, thereby supporting conclusions of previous studies that have linked spindle rotation and membrane invagination (e.g. Wang et al, 2011). In addition, antagonizing these inward forces exerted on the spindle, the authors propose the existence of attractive outward forces acting on the spindle. The authors do not provide direct evidence for the existence of these outward forces, nor their underlying mechanisms. Instead the authors use mathematical modelling predicting the existence of these forces to reconstitute aspects of spindle rotation. Overall, a thorough study, which requires some more direct evidence to support the authors' hypotheses.

- In Figure 2, the authors use a PKL1 inhibitor to block cytokinesis in oocytes, which subsequently fail to undergo spindle rotation. However, spindles do not only fail to rotate, but also start spinning around along the cell periphery as mechanisms required to maintain the spindle in its position are inhibited. This leaves the question open whether spindle rotation fails because of cytokinetic failure or loss of anchorage mechanisms. It could be argued that loss of spindle position alone is sufficient to hinder the establishment of the cytokinetic machinery above the spindle. Thus to strengthen the authors' conclusion that cytokinetic ring assembly and cortical ingression drives spindle rotation could benefit from additional evidence. For example, could the authors target more directly components of the cytokinetic machinery such as Ect2, Rho or Rock-1, by using specific drugs or dominant-negative variants? It would be helpful if under these conditions the spindle remains in place and solely cytokinesis is perturbed.

- Fig. 5D: In oocytes with dominant-negative Cdc42 that are still able to undergo cytokinesis, despite a small polar body (group 1), still appear to rotate their spindle successfully and effectively. However, from the authors' initial data one would expect that a perturbed polar body extrusion should somewhat impact on spindle rotation. In these cells, is spindle rotation delayed?

- Fig. 5G: These data require more thorough explanation. The authors show that the inward-directed cytoplasmic flows and spindle off-centering in dominant-negative Cdc42 occur at the same speed. The authors then conclude that the spindle off-centering is because the Cdc42 creates counterforces at the spindle to resist this inward flow. Alternatively, one could speculate that the cytoplasmic streaming forces are increased in dominant-negative Cdc42 resulting in spindle off-centering. Thus, the authors should clarify whether the flow velocities have been measured in dominant-negative Cdc42 and/or in control oocytes. They also could provide data to compare the flow velocities in control and Cdc42 to strengthen their conclusion that it is not an increase in inward-directed flows that leads to spindle off-centering.

- In line with these experiments, the authors conclude on line 248/249 that their Cdc42 loss of function experiments showed that the polarized domain exerts attractive forces on the chromosomes. However, the existence of such forces can at best be suggested from the provided experiments but are not demonstrated. To strengthen their conclusion, could the authors provide direct evidence for such forces by acutely inhibiting key players (Arp2/3, WASP or Myosin-II) in anaphase II that are involved in attracting the spindle in metaphase II? Alternatively, overexpression of downstream factors such as Arp2/3, WASP etc. might rescue to the spindle off-centering defects in dominant-negative Cdc42 oocytes. The authors' conclusion would greatly benefit from such and other experiments that show more directly the existence of such attractive forces acting on the spindle.

- A previous study (Yi et al, 2011) showed that upon loss of Arp2/3 the direction of cytoplasmic streaming is reversed in metaphase II and that this inversion might depend on Myosin-II function. Thus it is logical to speculate that Myosin-II might also be responsible for the detected inward flows in anaphase II. Have the authors tested the effects of Myosin-II inhibition on cytoplasmic flows in control and dominant-negative Cdc42, as well as on spindle off-centering in dominant-negative Cdc42?

Minor points:

- Fig. 4C: It appears that only individual cells are represented in the graphs, but better to show mean of several cells, if possible. In fact, Fig, 4C suggests that the outward cluster becomes attracted to the cell periphery (and inward cluster repelled from the cell periphery) before the onset of spindle rotation (about 20 minutes). Could the authors comment on this asymmetry is achieved before spindle rotation?

- Line 236: Speaking of inversion of flows, is confusing here, as in the given context this implies that the flows are inverted from control to dominant-negative Cdc42 oocytes. If authors meant to say that the flows were inverted from metaphase II to anaphase II, this should be stated more clearly.

- Line 240, Fig. 5G: Can the authors specify how the average speed of cytoplasmic flows was measured - only in the central part of the cell or as an average of the entire cell? Also, it should be clearly stated from how many cells these were measured. It appears that in Fig. 5G, only one cell per condition was used to perform vector measurements. If possible, could authors provide an overlay of several cells e.g. in form of a heatmap?

---

## [Decision Letter · Decision Letter 2]

1 Jul 2021

Dear Dr Dehapiot,

Thank you for submitting your revised Research Article entitled "RhoA- and Cdc42-induced antagonistic forces underlie symmetry breaking and spindle rotation in mouse oocytes." for publication in PLOS Biology. I have now obtained advice from the three original reviewers and have discussed their comments with the Academic Editor. 

Based on the reviews (attached below), we will probably accept this manuscript for publication, provided you satisfactorily address the remaining points raised by Reviewer 2. Please also make sure to address the data and other policy-related requests indicated below.

We expect to receive your revised manuscript within two weeks. 

*Published Peer Review History*

*Early Version*

Sincerely,

Ines

--

Ines Alvarez-Garcia, PhD,

Senior Editor,

PLOS Biology

Fig. 1D-H; Fig. 2D, F, G; Fig. 3B, D; Fig. 4B, C; Fig. 5B, C; Fig. 6C, D; Fig. 7B, D-G; Fig. S1 D, F; Fig. S3C; Fig. S4B, D; Fig. S5B-D and Fig. S6A-C

BLURB

Please also provide a blurb which (if accepted) will be included in our weekly and monthly Electronic Table of Contents, sent out to readers of PLOS Biology, and may be used to promote your article in social media. The blurb should be about 30-40 words long and is subject to editorial changes. It should, without exaggeration, entice people to read your manuscript. It should not be redundant with the title and should not contain acronyms or abbreviations. For examples, view our author guidelines: https://journals.plos.org/plosbiology/s/revising-your-manuscript#loc-blurb

Reviewers’ comments

Rev. 1:

The authors answered all my comments, hence I am fully satisfied with the revised work. The authors can be very proud of their work, it is a very nice and well conducted study, that I enjoyed reviewing. Congratulations to them!

Rev. 2:

The authors have done an excellent job in improving the manuscript based on previous reviewer comments. The low nocodazole experiment demonstrating that a cortical attractive force on chromosome clusters still exists during anaphase II is a nice addition. The work is a significant advance over previous work. However, there are two related issues that frustrate this reviewer and would frustrate any reader. First, the mechanism of the attractive force has not been determined. It is implied that cytoplasmic flow toward the cortex is the attractive force. The authors call this metaphase flow that persists during anaphase II. This leads to the second frustrating point. If cytoplasmic flow toward the cortex is the attractive force on both chromosome clusters, two sets of flow toward the cortex under each chromosome cluster should always be apparent in controls during anaphase II. This pair of flows toward the cortex is apparent in the averaged PIV figures (Fig. 2H and 6F) but not in the PIV figures of single movies (Fig. 1C and 5E). If the authors wish to state that metaphase flow to the cortex might be the attractive force, they should clarify how often flow toward the cortex under both chromosome clusters is observed in controls. If the authors think that metaphase flow is flow of the cytoplasm, flow cannot occur in opposite directions in the same place at the same time. If the authors think that metaphase flow is something different than the flow of the cytoplasm, they should explain this in the discussion.

Rev. 3: Binyam Mogessie – note this reviewer has signed his review

The authors have done a tremendous amount of work in response to all reviewer comments and returned with a much more robust manuscript with a stronger body of evidence to support their overall conclusions. I probably would have not attempted imaging spindle rotation in IVF eggs in reponse to reviewer 2's comment 6 as this was not going to be easy from the outset, as the authors found out. Perhaps a SrCl2 activation would have done the job. Their effort is admirable nonetheless and now we at least have it on record someone tested this approach. I am overall satisfied with the revised manuscript and fully support its publication.

---

## [Editor Report · Decision Letter 3]

30 Jul 2021

Dear Dr Dehapiot,

On behalf of my colleagues and the Academic Editor, Rebecca Heald, I am pleased to say that we can in principle offer to publish your Research Article entitled "RhoA- and Cdc42-induced antagonistic forces underlie symmetry breaking and spindle rotation in mouse oocytes." in PLOS Biology, provided you address any remaining formatting and reporting issues. These will be detailed in an email that will follow this letter and that you will usually receive within 2-3 business days, during which time no action is required from you. Please note that we will not be able to formally accept your manuscript and schedule it for publication until you have made the required changes.

PRESS

Sincerely, 

Ines

--

Ines Alvarez-Garcia, PhD 

Senior Editor 

PLOS Biology
